

# A new Digital Elevation Model of Antarctica derived from CryoSat-2 altimetry

Thomas Slater[1], Andrew Shepherd[1], Malcolm McMillan[1], Alan Muir[2], Lin Gilbert[2], Anna E. Hogg[1], Hannes Konrad[1], Tommaso Parrinello[3]

[1]Centre for Polar Observation and Modelling, School of Earth and Environment, University of Leeds, Leeds, LS2 9JT, United Kingdom
[2]Centre for Polar Observation and Modelling, University College London, London, WC1E 6BT, United Kingdom
[3]ESA ESRIN, Via Galileo Galilei, 00044 Frascati RM, Italy

*Correspondence to*: Thomas Slater (py10ts@leeds.ac.uk)

**Abstract.**

We present a new Digital Elevation Model (DEM) of the Antarctic ice sheet and ice shelves based on $2.5 \times 10^8$ observations recorded by the CryoSat-2 satellite radar altimeter between July 2010 and July 2016. The DEM is formed from spatio-temporal fits to elevation measurements accumulated within 1, 2 and 5 km grid cells, and is posted at the modal resolution of 1 km. Altogether, 94 % of the grounded ice sheet and 98 % of the floating ice shelves are observed, and the remaining grid cells

North of 88 º S are interpolated using ordinary kriging. The median and root mean square difference between the DEM and $2.3 \times 10^7$ airborne laser altimeter measurements acquired during NASA Operation IceBridge campaigns are -0.30 m and 13.50 m, respectively. The DEM uncertainty rises in regions of high slope — especially where elevation measurements were acquired in Low Resolution Mode — and, taking this into account, we estimate the average accuracy to be 9.5 m — a value that is comparable to or better than that of other models derived from satellite radar and laser altimetry.

**1 Introduction**

Digital Elevation Models (DEMs) of Antarctica are important datasets required for the planning of fieldwork, numerical ice sheet modelling, and the tracking of ice motion. Measurements of ice sheet topography are needed as a boundary condition for numerical projections of ice dynamics and potential sea level contributions (Cornford et al., 2015; Ritz et al., 2015). Accurate knowledge of surface elevation is also essential for both the delineation of drainage basins and estimation of grounding line

ice thickness, necessary for estimates of Antarctic mass balance calculated via the mass budget method (Rignot et al., 2011b; Shepherd et al., 2012; Sutterly et al., 2014). Furthermore, detailed and up-to-date DEMs are required to distinguish between phase differences caused by topography and ice motion when estimating ice velocity using interferometric synthetic aperture radar (Rignot et al., 2011a; Mouginot et al., 2012).



Previously published DEMs of Antarctica have been derived from satellite radar altimetry (Helm et al., 2014, Fei et al., 2017), laser altimetry (DiMarzio et al., 2007), a combination of both radar and laser altimetry (Bamber et al., 2009; Griggs and Bamber, 2009), and through the integration of several sources of remote sensing and cartographic data (Liu et al., 2001; Fretwell et al., 2013). In addition, high resolution regional DEMs of the marginal areas of the ice sheet have been generated

from stereoscopic (Korona et al., 2009) and radiometer surveys (Cook et al., 2012). However, the accuracy of both of these remote sensing techniques is considerably reduced in ice covered areas.

CryoSat-2, launched in 2010, is specifically designed to overcome the challenges of performing pulse-limited altimetry over Earth's polar regions. With a high inclination, drifting orbit, and novel instrumentation which exploits interferometry to obtain

high spatial resolution measurements in areas of steep terrain, CryoSat-2 provides a high density network of elevation measurements up to latitudes of 88 $^{\circ}$ (Wingham et al., 2006). Here, we utilise a 6 year time series of elevation measurements acquired by CryoSat-2 between July 2010 and July 2016 to derive a comprehensive and contemporary DEM of Antarctica at a spatial resolution of 1 km, with high data coverage in both the ice sheet interior and its complex marginal areas. We then evaluate the accuracy of the generated DEM against a set of contemporaneous airborne laser altimeter measurements, obtained

during NASA Operation IceBridge campaigns, in several locations covering Antarctica's ice sheet and ice shelves.

## 2 Data and methods

### 2.1 CryoSat-2 elevation measurements

We use 6 years of CryoSat-2 Baseline-C Level 2 measurements of surface elevation recorded by the SIRAL (SAR Interferometer Radar Altimeter) instrument, mounted on the CryoSat-2 satellite, between July 2010 and July 2016. Over

Antarctica, SIRAL samples the surface in two operating modes: Low Resolution Mode (LRM) and Synthetic Aperture Radar Interferometric mode (SARIn). In LRM, Cryosat-2 operates as a conventional pulse-limited altimeter (Wingham and Wallis, 2010), illuminating an area of approximately 2.2 km$^2$, with an across track width of roughly 1.5 km. LRM is used in the interior of the ice sheet, where low slopes and homogenous topography on the footprint scale are generally well suited for pulse-limited altimetry.

In SARIn, SIRAL uses two receive antennae to perform interferometry, allowing the location of the point of closest approach (POCA) to be precisely determined in the across-track plane (Wingham et al., 2004). In SARIn mode, bursts of 64 pulses are emitted at a high Pulse Repetition Frequency, and Doppler processing is then used to reduce the along-track footprint to approximately 300 m (Wingham et al., 2006). This increased sampling density, and ability to calculate the along- and across-

track location of the POCA, make SARIn well suited for measuring the steep and complex topography found in the ice sheet margins.





The CryoSat-2 Level 2 elevation product has a series of geophysical corrections applied to correct the selected measurements for the following: off-nadir ranging due to slope, dry atmospheric propagation, wet atmospheric propagation, ionosphere propagation, solid earth tide and ocean loading tide (ESA, 2012). For the ice shelves, additional inverse barometric and ocean tide corrections are also applied. Within the LRM mode mask area we use the Offset Centre of Gravity (OCOG) retracking

algorithm to retrieve the L2 elevation estimates, which defines a rectangular box around the centre of gravity of an altimeter waveform based upon its power distribution (Wingham et al., 1986). The OCOG retracking point is taken to be the point on the leading edge of the waveform which first exceeds 30 % of the rectangle's amplitude (Davis, 1997). We select the OCOG retracker as it offers robust retracking over a wide range of surfaces, and is adaptable to a variety of pulse shapes (Wingham et al., 1986; Davis, 1997). For the SARIn area we use the ESA L2 SARIn retracker, which determines the retracking correction

from fitting the measured waveform to a modelled SAR waveform (Wingham et al., 2006; ESA, 2012). Over the ice sheet and ice shelves, we use approximately 2.5 x $10^8$ CryoSat-2 elevation measurements to derive the new DEM.

## 2.2 DEM generation

To compute elevation, we separate the input CryoSat-2 elevation measurements into approximately 1.4 x $10^7$ regularly spaced 1 $km^2$ geographical regions. We then use a model fit method to separate the various contributions to the measured elevation

fluctuations within each region (Flament and Remy, 2012; McMillan et al., 2014). This method best suits CryoSat-2's 369-day orbit cycle, which samples along a dense network of ground tracks with few coincident repeats. We model the elevation $Z$, Eq. (1), as a quadratic function of local surface terrain $(x,y)$, a time invariant term $h$ accounting for anisotropy in radar penetration depth depending on satellite direction (Armitage et al., 2013), and a linear rate of elevation change $t$. The satellite heading term, $h$, is a binary term set to 0 or 1 for an ascending or descending pass, respectively.

$$Z(x,y,t,h) = \bar{z} + a_0x + a_1y + a_2x^2 + a_3y^2 + a_4xy + a_5(h) + a_6t \qquad (1)$$

We retrieve the model coefficients in each grid cell using an iterative least-squares fit to the observations to minimise the impact of outliers, and discard unrealistic estimates resulting from poorly constrained model fits. At a spatial resolution of 1 km this approach provides, on average, in excess of 30 elevation measurements per grid cell to constrain each solution. By using the model fit method, we are able to generate elevation estimates from 6 years of continuous Cryosat-2 data, which are

not unduly affected by fluctuations in surface elevation that may occur during the acquisition period (McMillan et al., 2014). In addition, it also allows for the retrieval of ice sheet elevation and rate of elevation change from the same data in a self-consistent manner.

We form the DEM from the mean elevation term, $\bar{z}$, in Eq. (1) within each 1 x 1 km grid cell. At a resolution of 1 km, the

model fit provides an elevation estimate in 60 % of grid cells within the ice sheet, and 75 % in the ice shelves. To fill data gaps in the 1 km grid, we generate additional DEMs of Antarctica from model fits at spatial resolutions of 2 km and 5 km. At these coarser resolutions more data are available to constrain model fits within a given geographical region, particularly at lower





latitudes where the spacing between ground tracks is larger. As a result, over the ice sheet the data coverage for DEMs generated at resolutions of 2 km and 5 km is increased to 91 % and 94 %, respectively (Fig. 1). For the ice shelves we use an additional DEM generated from model fits at a resolution of 2 km, for which data coverage is increased to 98 %. Data gaps in the 1 km grid are filled by the re-sampled 2 km and 5 km DEMs (where neither 1 km or 2 km model fit estimates are available)

for the ice sheet, and the 2 km DEM for the ice shelves (Fig. 2). This approach provides a DEM at the modal spatial resolution of 1 km, where approximately 94 % and 98 % of grid cells contain an elevation estimate derived from CryoSat-2 measurements for the ice sheet and ice shelves, respectively.

In order to provide a continuous dataset, we estimate elevation values in grid cells North of 88 ° S that contain no data using

ordinary kriging (Isaaks and Srivastava, 1989; Kitanitis 1997), with a search radius of 10, 25 or 50 km, depending on which first satisfies a minimum threshold of 100 data points to be used in the interpolation. Over the grounded ice sheet, 44 %, 52 % and 4 % of interpolated elevation values used a search radius of 10, 25 and 50 km, respectively. The majority of data points requiring a search radius of 50 km are located along the margins of Graham Land and Palmer Land in the Antarctic Peninsula, where data coverage is poor. After interpolation, the DEM provides a continuous elevation dataset for the ice shelves and ice

sheet for latitudes north of 88 ° S. We have chosen not to interpolate the pole hole due to interpolation distances exceeding the maximum kriging search radius of 50 km, and a desire to keep the DEM a product of CryoSat-2 data only.

**2.3 Airborne elevation measurements**

To evaluate the accuracy of the DEM, we compare our elevation estimates to measurements acquired by airborne laser altimeters during NASA's Operation IceBridge survey. The IceBridge mission, running since 2009, is the largest airborne

polar survey ever undertaken (Koenig et al., 2010). The primary goal of IceBridge is to maintain a continuous time series of laser altimetry over the Arctic and the Antarctic, bridging the gap between ICESat, which stopped collecting data in 2009, and ICESat-2, planned for launch in 2018.

We compare the DEM to elevation measurements obtained by two airborne laser altimeter instruments (Fig. 3):

•    The Airborne Topographic Mapper (ATM), over the following regions of the continental ice sheet: Antarctic Peninsula, Bellingshausen, Amundsen and Getz sectors of West Antarctica, and the Transantarctic Mountains, Oates Land and the plateau region of East Antarctica. The following ice shelves were also surveyed: Larsen C, Pine Island, Thwaites, Wilkins, Abbot, Getz, George VI, Ross and Filchner-Ronne. Measurements were acquired between March 2009 and December 2014 (Krabill, 2016).

•    The Riegl Laser Altimeter (RLA), over the Antarctic Peninsula, Marie Byrd Land of West Antarctica, Dronning Maud Land, Totten Glacier and Wilkes Land of East Antarctica, and the Ross ice shelf. Measurements were acquired between December 2008 and January 2013 (Blankenship et al., 2013).



The ATM is an airborne scanning laser altimeter capable of measuring surface elevation with an accuracy of 10 cm or better (Krabill et al 2004). Flown at a typical altitude of 500 m above ground level, the ATM illuminates a swath width of approximately 140 m, with a footprint size of 1-3 m and along track separation of 2 m (Levinsen et al., 2013). Data acquired by the RLA were collected as part of the NASA ICECAP program from December 2009 to 2013, mounted to a survey aircraft flown at a typical height of 800 m. Elevation measurements are provided at a spatial resolution of 25 m along track and 1 m across track with an error of approximately 12 cm (Blankenship et al., 2013).

In total, we selected approximately $2.3 \times 10^7$ laser altimeter elevation measurements, comprising of $1.7 \times 10^7$ ATM measurements, and $0.6 \times 10^7$ RLA measurements. Combined these data provide an independent comparison dataset, obtained over a contemporaneous time period and in a wide range of locations across Antarctica. For all airborne measurements a filter was applied to remove any erroneous step changes in elevation resulting from the laser altimeter ranging from cloud cover (Young et al., 2008; Kwok et al., 2012).

**2.4 DEM evaluation**

When comparing the DEM and airborne laser altimeter datasets, we separate the evaluation results according to whether the IceBridge elevation measurement resides in a grid cell derived from CryoSat-2 surface height measurements, hereby referred to as an observed grid cell, or an interpolated elevation value. This approach allows the accuracy of CryoSat-2 observations and the chosen interpolation method to be assessed independently. In total, approximately 84 % of the airborne laser elevation measurements reside within an observed DEM grid cell. Of this total, 53 %, 41 % and 6 % grid cells are derived from 1, 2 and 5 km model fits, respectively.

In order to compare the DEM and IceBridge datasets, we estimate elevations at the exact location of the airborne laser altimeter measurement through bilinear interpolation. Subsequently, we subtract the IceBridge elevation from the interpolated DEM elevation and collate the elevation differences into the same 1 x 1 km grid that the DEM is projected on. We then calculate the median difference to obtain one elevation difference for each individual grid cell, and to minimise the impact of outliers. On average, 1 km DEM grid cells overflown by IceBridge campaigns contain 70 individual airborne measurements. In total, elevation differences were compared for approximately $2.7 \times 10^5$ DEM grid cells, covering 2 % of the total ice sheet and ice shelf area, respectively. All DEM and IceBridge elevations are referenced to the WGS84 ellipsoid.

**3 Results**

Our new DEM of Antarctica (Fig. 4) provides an elevation value derived from CryoSat-2 measurements for 94 % of the grounded ice sheet and 98 % of the ice shelves. The remaining 5 % of grid cells North of 88 ° S are interpolated using ordinary kriging to provide a continuous gridded elevation dataset for the entire continent beyond the pole hole. Accounting for the



length of the elevation time series within each individual grid cell, we determine the effective time stamp of the DEM to be July 2013. Surface slopes derived from the DEM (Fig. 5) illustrate the short scale topographic undulations, and identify the ice divides and larger features such as subglacial Lake Vostok.

To evaluate the DEM's systematic bias we compute the median elevation difference with respect to the airborne measurements, as this is robust against the effect of outliers. To evaluate its random error we compute the root mean square (RMS) difference. Both of these statistical measures are more appropriate than the mean and standard deviation when describing the systematic bias and random error, respectively, of the non-Gaussian distributions we typically find when calculating elevation differences between the DEM and IceBridge elevation datasets.

**3.1 Comparison of DEM to airborne elevation measurements: observed grid cells**

A primary objective of NASA's IceBridge program is to maintain a continuous observational record of rapidly changing areas in Antarctica. As a result, elevation measurements were obtained in regions such as Pine Island (PIG), Thwaites and Totten Glaciers, where the observed thinning rate is of the order of several metres per year (McMillan et al., 2014). Therefore, we expect to see height differences between the DEM and airborne datasets due to real changes in surface elevation between their
respective acquisition periods. Comparing DEM elevations at PIG against measurements acquired by ATM flights in the years 2009, 2011 and 2014 (Fig. 6), the elevation difference is smallest in 2014, closest to the DEM effective date of July 2013 (Table 1).

To account for the temporal difference between the two datasets, we adjust the interpolated DEM value for changes in surface
elevation which may have occurred between the acquisition periods. We calculate this adjustment by interpolating the gridded elevation trends (Eq. (1)) to the location of the airborne measurement, through the same bilinear interpolation method as used for the DEM elevation estimate. The elevation change trends were corrected for temporal fluctuations in backscattered power, which can introduce spurious signals in time series of elevation change (Davis and Ferguson, 2004; Khvorostovsky, 2012).

Within the grounded ice sheet, we note that the spatial distribution of the airborne dataset used for comparison (Fig. 7) preferentially samples regions of high slope and low elevation, and does not reflect the overall distributions of the Antarctic ice sheet. Approximately 60 % of DEM grid cells overflown by IceBridge survey craft have an elevation of less than or equal to 1000 m, and 43 % have a surface slope greater than 0.5 °. In comparison, approximately 15 % and 22 % of the Antarctic ice sheet area has elevations of less than 1000 m and slopes greater than 0.5 °, respectively. At the continental scale, there is
generally good agreement between the DEM and airborne laser altimeter measurements (Fig. 8), and the median and RMS elevation difference between the DEM and airborne data are -0.27 m and 13.36 m., respectively (Table 2).





At the Antarctic Peninsula, the median and RMS difference are -1.12 m and 22.40 m, respectively; errors are larger in this region due to its mountainous and highly variable terrain, and it remains a challenge for radar altimetry. The largest elevation differences in this region are found in DEM grid cells derived from 5 km model fits, indicating that the complex topography is poorly described by a quadratic model. In grid cells with elevation values derived from 1 km model fits, which accounts for

40 % of the Antarctic Peninsula DEM, the median and RMS difference are improved to -0.71 m and 16.88 m, respectively. Geographically, elevation differences rise towards Graham Land at the northern tip of the Antarctic Peninsula, where topography is complex and highly variable at length scales similar to the satellite footprint.

In West Antarctica there is good agreement between the DEM and airborne measurements, particularly along the coastal

margins of the Bellingshausen and Amundsen Seas. In the Bryan and Eights Coasts in the Bellingshausen Sea Sector, the median and RMS difference are -1.72 m and 10.40 m, respectively. At Pine Island and Thwaites Glaciers, and their surrounding drainage area, the median difference is -1.02 m, and the RMS difference is 10.58 m. Further inland towards Marie Byrd Land, the median and RMS differences are 0.20 m and 5.27 m, respectively.

In East Antarctica, the DEM compares well to the airborne dataset inland in the plateau region where slopes are low, and the topography is well suited to satellite radar altimetry. Along the George coast and in George V Land, the median difference is -0.68 m, and the RMS difference 6.52 m. Over Totten glacier and its catchment area, the median and RMS difference are -0.39 m and 16.15 m respectively. In this region, there is good agreement with airborne elevations both inland towards Dome C, and over Totten glacier itself. On the eastern flank of Law Dome, biases of several tens of metres between the DEM and

the airborne data coincide with grid cells derived from 5 km model fits, where there is insufficient data to constrain models at higher spatial resolutions. As a result, elevations derived from 5 km model fits will poorly sample the highly sloping terrain in this region when compared to the airborne laser. Additionally in East Antarctica, elevation differences several tens of metres in magnitude over the Pensacola Mountains occur where high surface slopes and nunataks complicate radar altimeter elevation retrievals.

At the Antarctic ice shelves, the DEM also compares favourably to the airborne elevation data, with median and RMS differences of -0.42 m and 14.31 m, respectively. Differences are most pronounced near to grounding lines where tidal effects are relatively large and where the terrain is generally more complex, and are smallest in the interior of the larger ice shelves which are generally flat. At the Ross and Filchner-Ronne Ice Shelves, for example, the RMS differences are 3.93 and 3.54 m,

respectively — considerably lower than the continental average.

Overall, the median and RMS differences between the DEM and airborne measurements are -0.30 m and 13.50 m, respectively, and 99 % of the data agree to within 45 m. In addition to temporal mismatch, possible explanations for residual elevation differences include differences in the satellite and airborne altimeter footprint sizes and scattering horizons, as well as errors




in the individual data sets themselves. Although generally small, biases between the DEM and the airborne data are notably high in several isolated regions, including the upstream catchments of the Byrd Glacier flowing from East Antarctica into the Ross Ice Shelf, the Recovery Glacier flowing from East Antarctica into the Filchner Ice Shelf, and the Foundation Ice Stream in the Pensacola mountains (see Fig. 7). In each of these locations, surface slopes are high (exceeding 1 °) and CryoSat-2

operates in Low Resolution Mode (see Fig. 5). To illustrate this in more detail, we compare elevation recorded along two RLA tracks falling within the LRM zone (Fig. 9); one at Byrd Glacier where slopes are high and undulating, and another 600 km northward in Victoria Land where slopes are low and smooth. Along these tracks, elevation differences of approximately 20 m occur where the terrain undulates rapidly, because CryoSat-2 oversamples the topography when operating in LRM. Despite being well sampled by the airborne laser altimeter dataset, regions of high surface slopes represent a small fraction of the area

surveyed by CryoSat-2 in either LRM or SARIn modes (Table 3). In contrast, agreement between DEM and IceBridge elevations in regions of lower surface slope ($< 0.5$ °) — which represent the majority of the ice sheet — falls typically in the range 5 to 10 m in either operating mode (Table 3). Combining the slope-dependent errors (Table 3) and the distribution of slopes within the LRM and SARIn mode masks, we estimate the average uncertainty of the observed DEM to be 9.5 m.

## 3.2 Comparison of DEM to airborne elevation measurements: interpolated grid cells

A small proportion (5 %) of the DEM is estimated by ordinary kriging, and we assess the accuracy of this method by comparing airborne elevation measurements residing in a DEM grid cell containing no data with the interpolated value (Table 2). Predictably, our interpolated DEM values deviate more from the airborne elevation measurements in areas of high slope and complex terrain, where internal tracker losses occur and data coverage is reduced. This is true in particular for the ice sheet

margins and the Antarctic Peninsula, where there is little spatial correlation over the 10, 25 and 50 km search distances we have chosen for the interpolation, and limited data coverage available for sampling. At the Antarctic Peninsula, where the majority of interpolated grid cells are located in the bare rock regions on the north coasts of Graham and Palmer Land, the median and RMS difference are 82.21 m and 191.07 m, respectively. Similarly in East Antarctica, where the median and RMS difference are 19.62 m and 117.77 m, respectively, interpolated grid cells are primarily found in the rugged, bare rock terrain

across the Transantarctic Mountains, the Victory Mountains in Victoria Land, and the mountain ranges in Oates Land.

The largest interpolation errors are located in empty grid cells at boundary of the ice sheet along the margins, as data gaps are filled through extrapolation from data inland rather than interpolation between known values. Over higher elevation regions with relatively smooth topography it is more reasonable to assume spatial correlation over interpolation distances of 10 to 50

km, and our chosen interpolation method is more reliable. Within the LRM zone, the median and RMS difference are 6.51 m and 41.70 m, respectively. Because we have not corrected for changes in elevation over time occurring between the acquisition periods of the two datasets within our comparison of interpolated grid cells, there is additional error in elevation differences which is not accounted for in regions where rates of elevation change are large.



## 3.3 Comparison of currently available DEMs

We compare the accuracy of the new CryoSat-2 DEM over the ice shelves, Antarctic Peninsula, West Antarctica and East Antarctica with three other publically available Antarctic DEMs: Bedmap2 (Fretwell et al., 2013), and DEMs generated from ERS-1 and ICESat data (Bamber et al., 2009), and CryoSat-2 data (Helm et al., 2014). To ensure an equivalent comparison dataset, we only use airborne elevation measurements which reside in an observed grid cell of the presented CryoSat-2 DEM (see Fig. 7). For all four DEMs we use the same evaluation method as described in Sect. 2.4. From the calculated median and root mean squared differences, the new CryoSat-2 DEM we present here is comparable to, or an improvement upon currently available DEMs in all four regions (Fig. 8). In areas of high rates of elevation change, it is worth noting that all four DEMs will exhibit larger biases due to real changes in surface elevation between the acquisition periods of the respective datasets, and that these differences may be larger in the DEMs containing older ERS-1 and ICESat data (Bedmap2, Bamber et al., 2009). Although another recent DEM of Antarctica (Fei et al., 2017) formed using 1.7 x $10^7$ elevation measurements acquired by CryoSat-2 between 2012 and 2014 is not available for direct assessment, it has a reported accuracy of approximately 1 m for the high elevation region at the Domes, 4 m for the ice shelves and over 150 m for mountainous and coastal areas.

## 4 Conclusions

We present a new DEM of Antarctica derived from a spatio-temporal analysis of CryoSat-2 data acquired between July 2010 and July 2016. The DEM is posted at a modal resolution of 1 km and contains an elevation measurement in 94 % and 98 % of ice sheet and ice shelf grid cells, respectively; elevation in a further 5 % of the domain is estimated via ordinary kriging. We evaluate the accuracy of the DEM through comparison to an extensive independent set of airborne laser altimeter elevation measurements, acquired over a contemporaneous time period and in a wide range of locations across the Antarctic ice sheet and ice shelves. From a comparison at grid cells acquired in both data sets, the median and RMS difference between the DEM and airborne data are -0.30 m and 13.50 m, respectively. The largest elevation differences occur in areas of high slope and where CryoSat-2 operates in Low Resolution Mode, where the altimeter ranges to the peaks of undulating terrain and under samples troughs. Using the slope-dependent uncertainties and the wider distribution of slopes, we estimate the overall accuracy of the DEM to be 9.5 m where elevations are formed from satellite data alone. In areas where the DEM is interpolated, the median and RMS differences rise to 19.84 m and 131.13 m, respectively. Through comparisons to an equivalent validation dataset in four individual Antarctic regions, we find the new CryoSat-2 DEM to be comparable to, or an improvement upon, three publically available and widely used Antarctic DEMs.

## Acknowledgements

The new CryoSat-2 DEM will be made freely available to users via the Centre for Polar Observation and Modelling data portal (http://www.cpom.ucl.ac.uk/csopr/) and via the European Space Agency (ESA) CryoSat Operational Portal



(https://earth.esa.int/web/guest/missions/esa-operational-eo-missions/cryosat). This work was led by the NERC Centre for Polar Observation and Modelling, supported by the Natural Environment Research Council (NERC) (cpom300001), with the support of grant (4000107503/13/I-BG). We acknowledge ESA for the provision of CryoSat-2 data (available at https://earth.esa.int/web/guest/-/cryosat-products), ESA's Antarcic_Ice Sheet_cci, and the National Snow and Ice Data Center

for the provision of IceBridge airborne altimetry data (available at https://nsidc.org/icebridge/portal). We also acknowledge the authors of the three digital elevation models used in this study, all of which are freely available online. T.S is funded through the NERC iSTAR Programme and NERC grant number NE/J005681/1. A.E.H is funded from the European Space Agency's support to Science Element program, and an independent research fellowship (4000112797/15/I-SBo).

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



| Year | Number of compared grid cells | Median difference (m) | RMS difference (m) |
|------|------|------|------|
| 2009 | 3490 | -4.97 | 11.35 |
| 2011 | 2819 | -2.08 | 8.56 |
| 2014 | 2794 | 0.05 | 3.97 |

**Table 1: Statistics of the comparison between observed DEM grid cells derived from 1 km model fits, and ATM elevation measurements in the Pine Island Glacier drainage basin. The airborne data are separated into the year of acquisition to demonstrate the effect of ice dynamical thinning in this region on the elevation difference. The effective date of the DEM is July 2013.**




|  | **Observed** | | | **Interpolated** | | |
| Region | Number of compared grid cells | Median difference (m) | RMS difference (m) | Number of compared grid cells | Median difference (m) | RMS difference (m) |
|---|---|---|---|---|---|---|
| Ice sheet | 230165 | -0.27 | 13.36 | 32933 | 25.37 | 138.62 |
| Ice shelves | 40081 | -0.42 | 14.31 | 4772 | 1.20 | 30.96 |
| Peninsula | 6820 | -1.12 | 22.40 | 7473 | 82.21 | 191.07 |
| West Antarctica | 60452 | -0.86 | 11.43 | 8783 | 11.78 | 96.15 |
| East Antarctica | 162893 | -0.17 | 13.60 | 14679 | 19.62 | 117.77 |
| LRM | 73867 | 0.26 | 7.15 | 1683 | 6.51 | 41.70 |
| SARIn (ice sheet only) | 156298 | -0.82 | 15.45 | 31250 | 28.65 | 141.97 |
| Total | 270246 | -0.30 | 13.50 | 37655 | 19.84 | 131.13 |

**Table 2: Statistics of the comparison between observed and interpolated DEM grid cells and airborne elevation measurements for individual Antarctic regions and mode mask areas. In total, only 5% and 2% of DEM elevation values are obtained through interpolation for the ice sheet and ice shelves, respectively.**



| Slope (°) | LRM | | SARIn | |
| --- | --- | --- | --- | --- |
| | RMS difference (m) | LRM area coverage (km$^2$) | RMS difference (m) | SARIn area coverage (km$^2$) |
| 0-0.25 | 4.90 | 5481579 | 6.37 | 1373258 |
| 0.25-0.5 | 11.24 | 975624 | 8.54 | 1338826 |
| 0.5-0.75 | 19.85 | 143420 | 13.50 | 775083 |
| >0.75 | 29.59 | 93660 | 24.26 | 1551836 |

**Table 3: RMS differences between observed DEM grid cells airborne elevation measurements for four slope bands, separated into the LRM and SARIn mode mask areas for the Antarctic ice sheet. The area of each region represented by the four slope bands is also provided.**



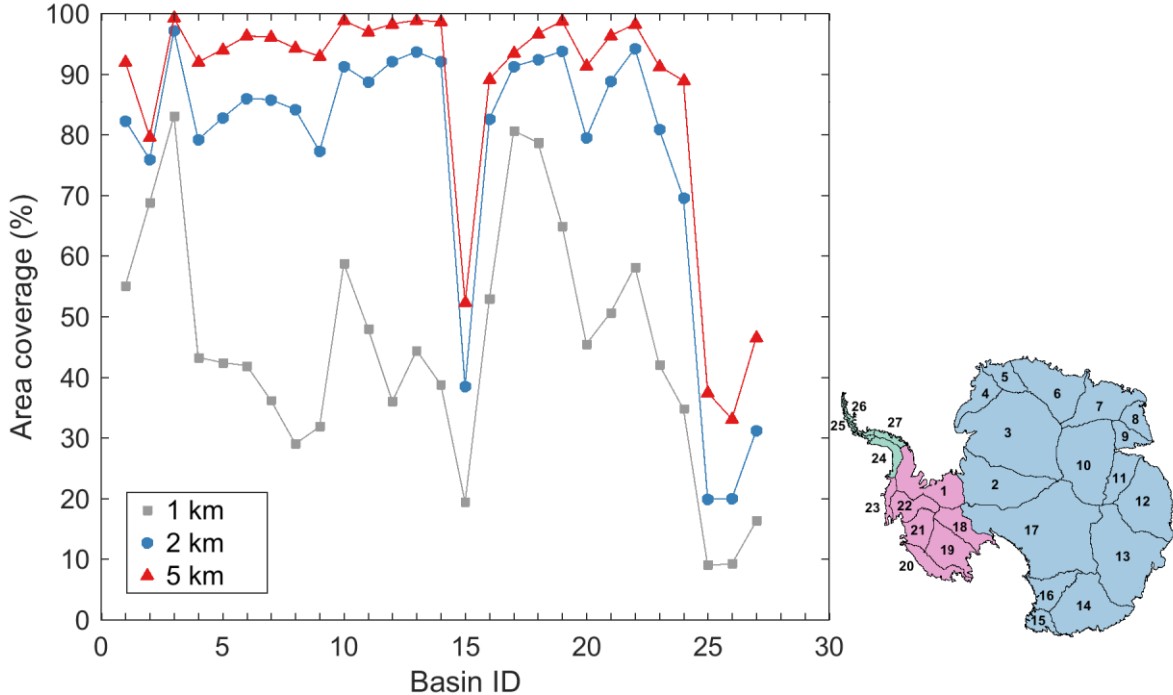

**Figure 1: Area coverage of elevation values provided by the model fit solution of CryoSat-2 measurements for the Antarctic ice sheet, with a grid cell sizing of 1 km², 2 km² and 5 km². Solid black lines and numbers (inset) show the boundaries and ID numbers of the 27 drainage basins used (Zwally et al., 2012). East Antarctica and the Antarctic Peninsula are defined as numbers 2 to 17 and 24 to 27, respectively, and the remaining numbers define West Antarctica.**



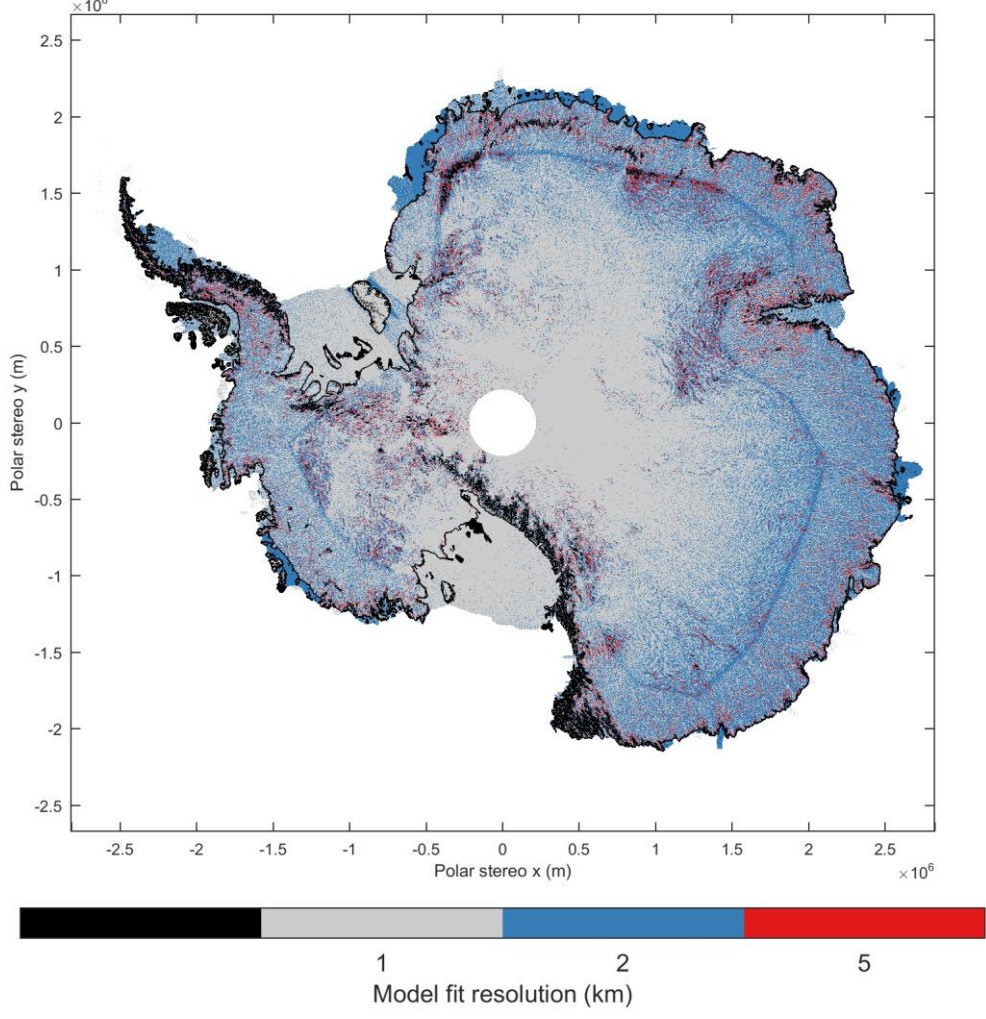

**Figure 2:** The grid cell resolution of the model fit method used to derive the surface elevation in each 1 km grid cell. Elevation values obtained from the 2 km and 5 km model fits are oversampled to the modal DEM resolution of 1 km. A black grid cell denotes a cell that contains an interpolated value. For the grounded ice sheet, approximately 60%, 30% and 5% of elevation values are derived from 1, 2 and 5 km model fits, respectively. For the ice shelves, 75% of elevations are calculated with 1 km model fits, and 23% from 2 km model fits. The remaining 5% of ice sheet and 2% of ice shelf values are interpolated using ordinary kriging.





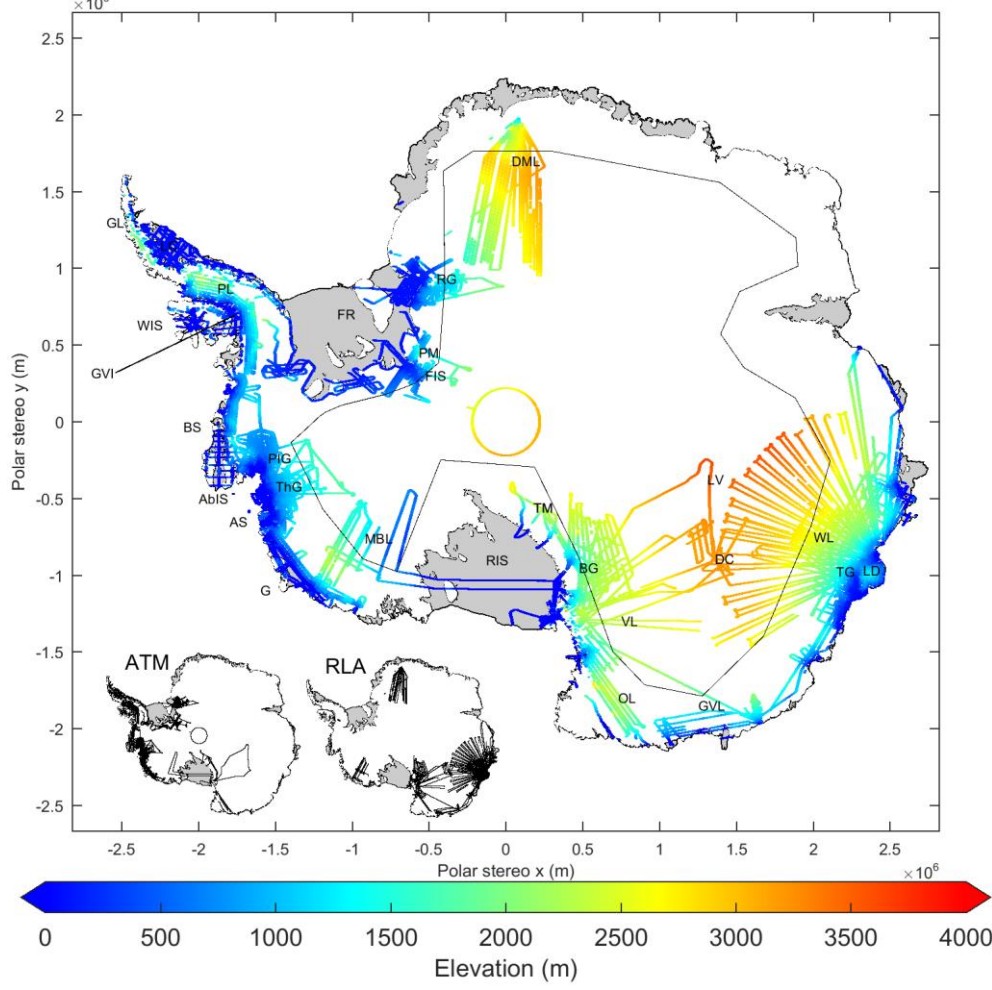

**Figure 3: IceBridge airborne dataset used to evaluate the DEM, acquired between December 2008 and December 2014. The mode mask boundary (black) between CryoSat-2 LRM and SARIn modes is also shown. (inset) Locations of the individual ATM and RLA airborne datasets. Labelled are the following locations of interest: AbIS: Abbot Ice Shelf, AS: Amundsen Sea, BS: Bellingshausen Sea, BG: Byrd Glacier, DC: Dome C, DML: Dronning Maud Land, FIS: Foundation Ice Stream, FR: Filchner-Ronne Ice Shelf, G: Getz, GL: Graham Land, GVI: George VI Ice Shelf, GVL: George V Land, LC: Larsen-C Ice Shelf, LD: Law Dome, LV: Lake Vostok, MBL: Marie Byrd Land, OL: Oates Land, PIG: Pine Island Glacier, PL: Palmer Land, PM: Pensacola Mountains, RG: Recovery Glacier, RIS: Ross Ice Shelf, TG: Totten Glacier, ThG: Thwaites Glacier, TM: Transantarctic Mountains, VL: Victoria Land, WIS: Wilkins Ice Shelf, WL: Wilkes Land.**





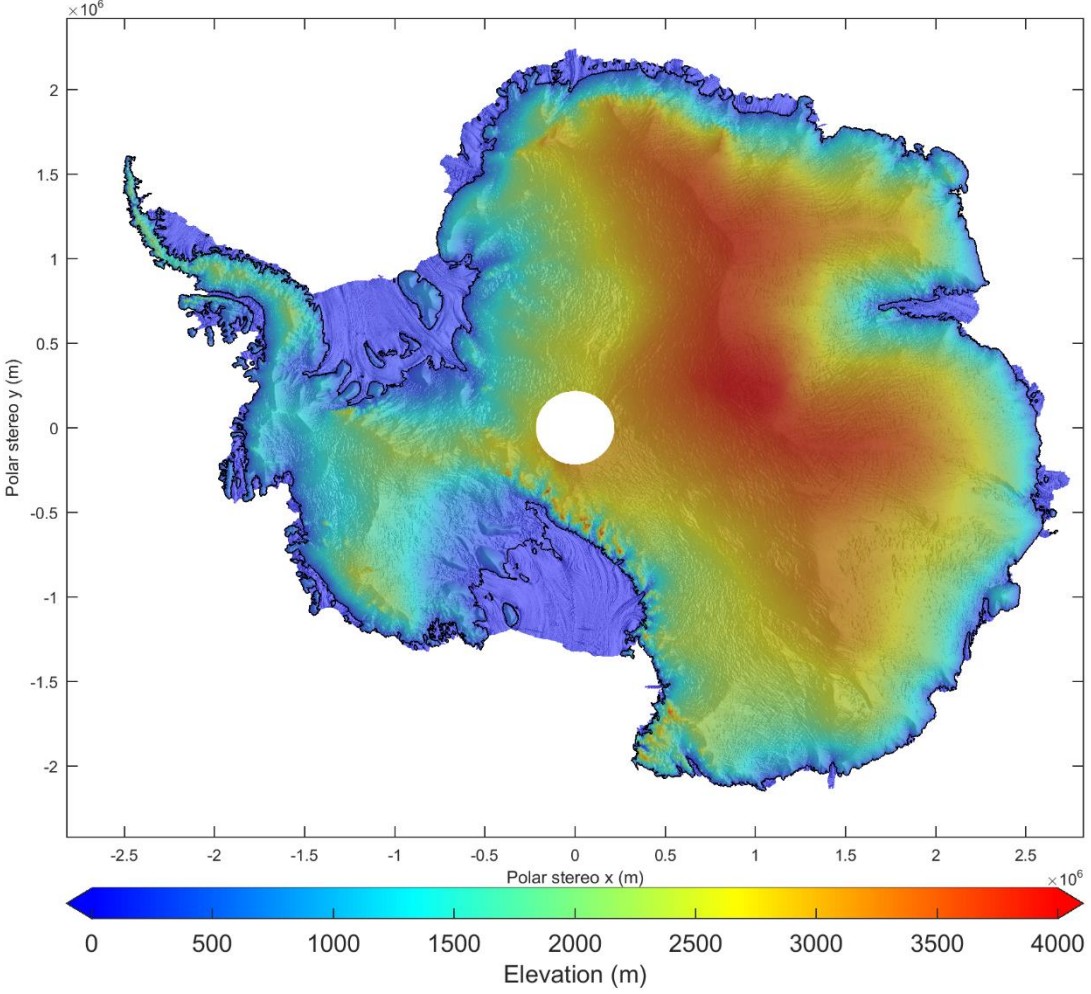

**Figure 4: A new elevation model of Antarctica derived from 6 years of CryoSat-2 radar altimetry.**



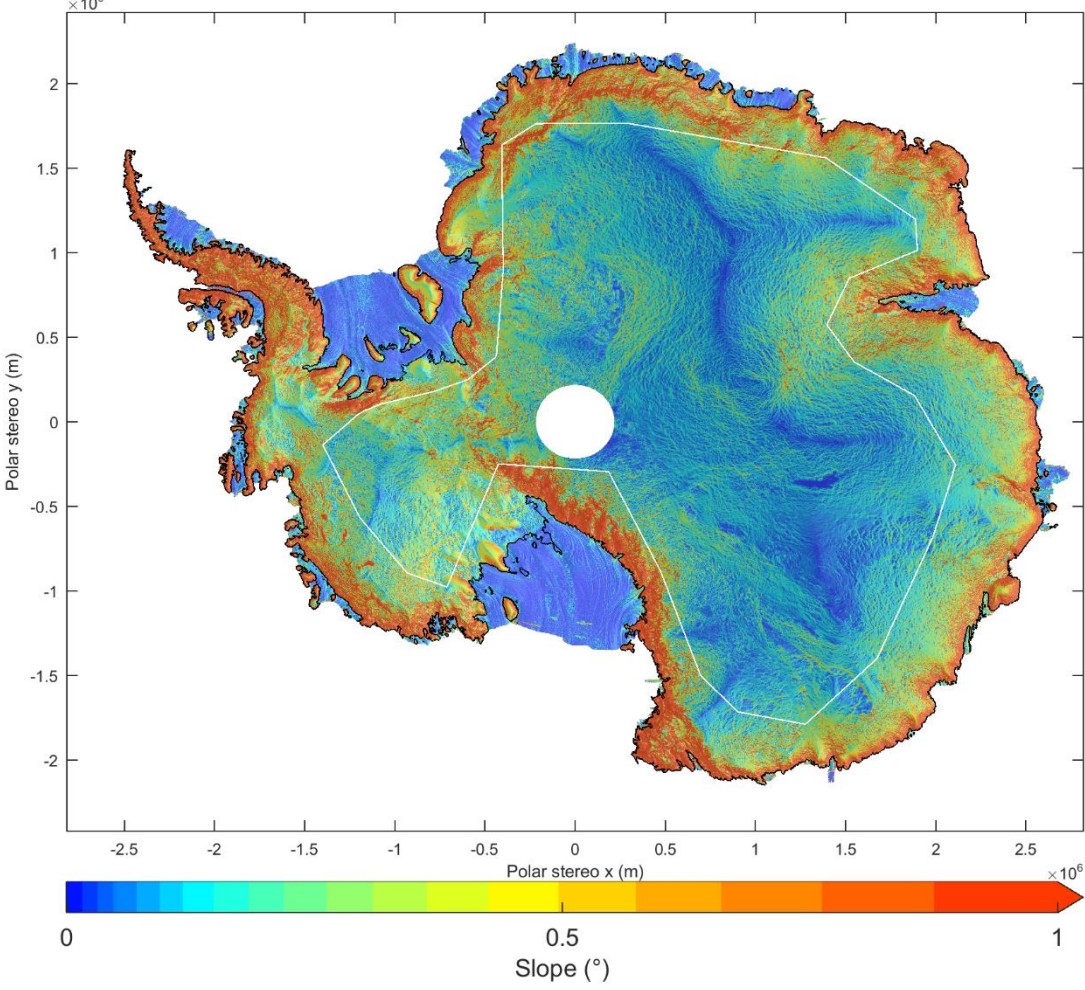

**Figure 5: Surface slopes of Antarctica posted at a resolution of 1 km, derived from the digital elevation model. The mode mask boundary between CryoSat-2 LRM and SARIn modes is also shown in white.**





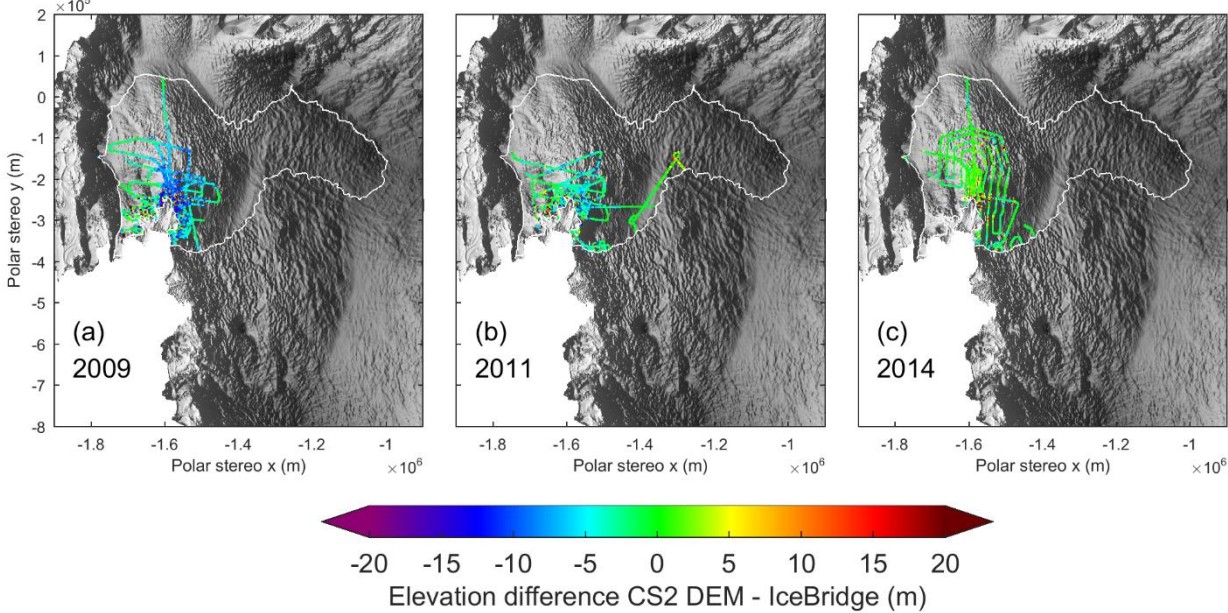

**Figure 6:** Difference between observed DEM grid cells derived from 1 km model fits and IceBridge ATM elevation measurements for the Pine Island Glacier region in West Antarctica for ATM flight surveys undertaken in the years (a) 2009, (b) 2011 and (c) 2014. The DEM has an effective time stamp of July 2013. The boundary of the PIG drainage basin (white) is also shown (Zwally et al., 2012).



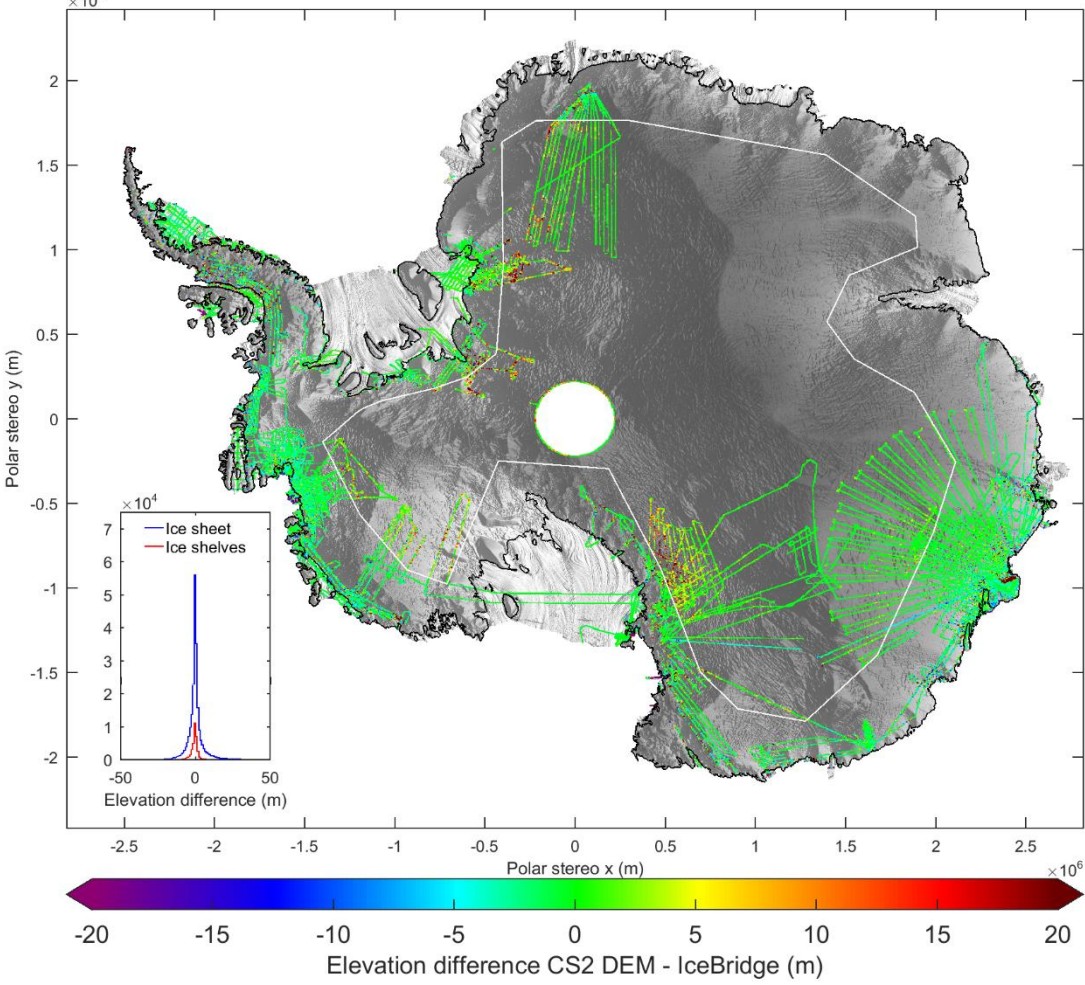

**Figure 7: Difference between CryoSat-2 DEM elevation and airborne laser altimeter measurements in observed grid cells. Elevation differences are overlaid on a shaded relief plot of the DEM. (inset) Distribution of elevation differences (DEM − airborne) for the ice sheet and ice shelves.**



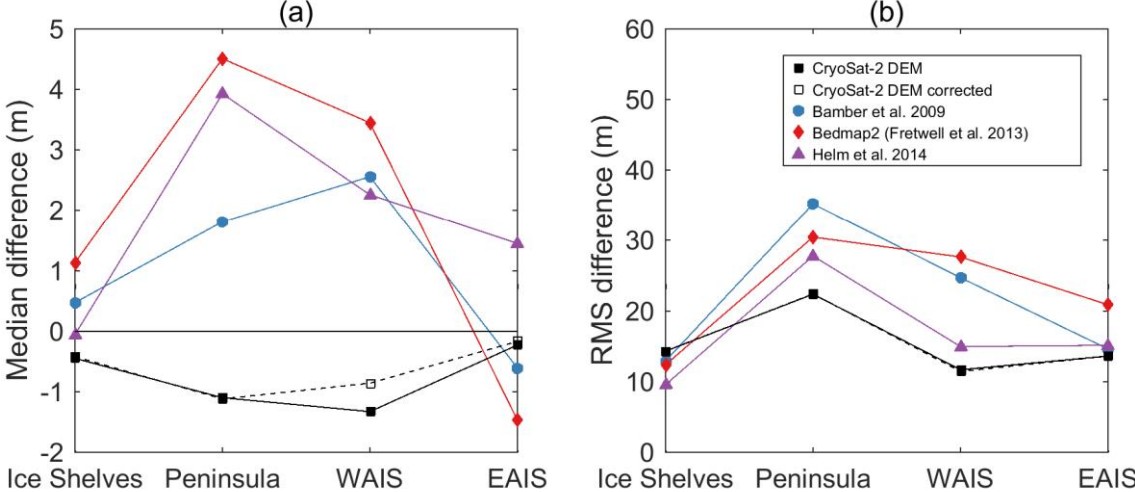

**Figure 8: (a)** Median and **(b)** RMS differences between airborne elevation measurements calculated over the ice shelves, Antarctic Peninsula, West Antarctica (WAIS) and East Antarctica (EAIS) for the new CryoSat-2 DEM presented in this report, and three publically available Antarctic DEMs. CryoSat-2 DEM comparisons with the elevation change correction applied (Table 2) are plotted as square outlines.







**Figure 9:** CryoSat-2 LRM, and IceBridge RLA elevation profiles for 100 km flight path sections obtained in (a) Victoria Land, where surface slopes are low, and (b) inland from Byrd Glacier, where surface slopes are high. Elevation differences (CS2 DEM – airborne) are plotted in blue to the right hand scale. (Inset) locations of RLA flight paths, with the profile section highlighted in red. The SARIN/LRM mode mask boundary is shown as a dashed line.