# Peer review of "A new Digital Elevation Model of Antarctica derived from CryoSat-2 altimetry"

_The Cryosphere, 2017_

## Referee Comment (RC1) · Anonymous Referee #1 · 8 Nov 2017

This is a generally well written and sound paper that describes the construction of a new Cryosat2 DEM and tests it accuracy. The paper shows that the new DEM exceeds the accuracy of other elevation models for the ice sheets and therefore it will be very useful to the Antarctic and Cryosphere community as a whole. Apart from a few small points I would suggest that the manuscript is good I list those minor points below: 1. Page 1 Line 24: "surface elevation is essential for delineation of drainage basins" – this is not totally true – although early work on drainage basins (early Rignot and Zwally papers), used surface slope to delineate drainage, later Rignot papers used flow and flow direction which is a more robust method. I would temper the sentence by saying "can be used" rather than are essential. 2. Page 2 line 5: It might be worth adding that over rock outcrops and steeper slopes, where cryostat coverage is

poorer, the photogrammetric models do perform better and so act as an alternative in these regions. 3. Page 3 line 30. Reading the text I was a little unsure if the figures (percentages) included the polar hole or not. Please clarify. 4. Page 4 line 10: Why use Kriging? There are lots of alternative interpolation methods, all will give slightly different results, but you do not say why this method was chosen rather than the others. Do you have any evidence that this is the best method to use? 5. A slight grammatical point on page 7 and through the text; the phase " At the Antarctic Peninsula" and "At the Antarctic Ice shelves" should be changed to "On the …..." 6. Page 9 line 22+23: are there also more data gaps in the high slope areas that add to the inaccuracies over these regions? If so it would be worth adding the point. 7. Figure 1 and figure 8: please remove the lines connecting the points as these are misrepresentative. Both these figures would be better as histograms rather than line graphs. 8. In the figure legend for figure 2 I would add a point explaining the noticeable blue line where the mode changes from LRM and SARIn modes.

---

## Referee Comment (RC2) · Anonymous Referee #2 · 25 Nov 2017

Title: A new Digital Elevation Model of Antarctica derived from CryoSat-2 altimetry
Author(s): Thomas Slater et al.

This paper presents a new digital elevation model of Antarctica based on 6 years of
CryoSat-2 measurements

To generate the DEM the authors use a different approach as most studies before.
Instead of applying an interpolation method like IWD or Kriging a function is fitted on a
pixel level to estimate elevation, 2D quadratic surface' and elevation trend at the same
time. Empty neighboring pixels are filled at a later stage using kriging interpolation.
The final DEM is validated against ICEBridge data and slightly compared to existing
DEMs.

[Figure]

The paper reads well and the figures and tables are illustrative and informative. Validation against Airborne data is well explained and scientifically sound.

However, to my opinion, since the paper is presenting another Antarctic DEM a more detailed analysis and comparison to existing DEMs is required. Especially the implications, reliability of the new method compared to widely used interpolation methods is worth to investigate in more depth. E.g. a comparison with the cited CryoSat-2 DEM of Helm et.al. could be used to demonstrate if and where this new approach in combination with 6 times more data is performing better or has weaknesses. E.g. a difference plot between both CryoSat-2 DEMs over whole Antarctica would be very informative to see in which areas the DEMs differ.

I have some concerns about the applied pixel fit method. Since the elevation is generated on a pixel level (neighboring pixel are not 'talking to each other'). This might introduce elevation jumps or artefacts. As example I generated hill shades of the new DEM and the mentioned external DEMs (Fig1). The new DEM shows clearly erroneous pixels, especially in areas of steep topography, close to the Grounding line. I'm wondering if the authors can explain why this is happening and if there is a strategy to avoid this. The other DEMs do not show such artefacts.

In table 3 a comparison of the Differences of LRM/SARIn areas to Icebridge with respect to slope is given. Since the new DEM is a composite of 'observed pixels' and 'interpolated pixels' I would like to see this analysis splitted up (similar to Table 2).

Furthermore, I would suggest to present two figures where the mean difference and respectively the Stddev is plotted against slope (e.g. binned to $0.05°$) for the whole DEM, observed and interpolated pixel, respectively. In addition, this Figure should include the same analysis for the other 3 DEMs. Such kind of figure would clearly show the difference between observed and interpolated pixels as well as the stated improvement of the new DEM against existing DEMs.

Is there a way to compare the final slope model (Fig 5) with the 2D quadratic surface

slope, estimated for each pixel?

How robust is this pixel wise 2D surface fit and its sensitivity to the number and distribution of data points within a pixel? One could run the same fit method on yearly subsets of data and compare the elevations of 'observed pixels' of the subset DEMs with the Final DEM. (e.g. generate maps of the mean of the differences between subset-DEMs and final DEM and its stddv).

Please make clear what kind of data you are using as input. You mention that an OCOG retracker is used for LRM data but another retracker for SARIn. Please explain why and what are the consequences (e.g bias based on the different retracker).Are you applying any slope correction to the LRM data? If yes what method is used and which DEM is used for the slope correction? Do you use your newly derived DEM and an iterative scheme? If you use another DEM, please explain why. If you don't apply a slope correction to LRM than I see some inconsistency between the LRM and SARIn regions, since the SARIN data is slope corrected using the interferometric phase. Do you use all data points or do you run a pre-filtering to exclude erroneous data points before running the fit procedure? Are there any filtering approaches used after applying the pixel wise fit?

I also miss a map of uncertainty for whole Antarctica coming along with the DEM which would be required for ice sheet modelers.

P8 L30: Why are you not applying an elevation trend correction to the interpolated areas? This is inconsistent. You could easily generate an interpolated dhdt map to correct for elevation change in interpolated areas.

Fig 8: Why is the median difference below zero in all cases - this would mean that CryoSat is measuring above the laser surface?

Figures (2,3,4,5,6,7,9): please use km instead of m and overplot a Latitude / Longitude grid on top of the polarstereo projection.

Furthermore, I think that the new DEM is shifted by 1 pixel. Difference plots as suggested above show a strange pattern (see Fig. 2) which is not observed between the other 3 DEMs - Do you have any explanation?

[Figure]

Bamber DEM

CS-DEM (2012)

New CS-DEM

**Fig. 1.** Hill shades of 3 DEMs. (New DEM Pixel errors e.g. Berkner Island

**Fig. 2.** Difference between IceSat DEM and New DEM (colour scale: -25 to +25)

---

## Author Comment (AC1) · 30 Nov 2017

**Preliminary response to Anonymous Referee #2**

Thomas Slater, Andrew Shepherd, Malcolm McMillan, Alan Muir, Lin Gilbert, Anna E. Hogg, Hannes Konrad, Tommaso Parrinello

We would like to thank you for your constructive review of the manuscript. We will address the rest of your comments in due course – however we would like to immediately address Fig. 2 included in your review. We have tried to reproduce your figure by plotting difference maps between the ICESat DEM, the 1 km CryoSat-2 DEM presented in this paper, and other publicly available DEMs (see Fig. 1 a) to c)). However, we cannot replicate the pattern you are experiencing, and see similar difference patterns for all tested DEMs. Please could you provide more information as to how Fig. 2 was produced? Specifically, please could you cite which input ICESat dataset was used (geoid, posted resolution, etc.)? Because the Antarctic ICESat DEM is posted at a resolution of 500 m, different to the new CryoSat-2 DEM, one of the DEMs must be resampled to the other to generate the difference plot. Please could you specify which method was used so we can keep our method consistent with the one used in your review?

In addition, can we please ask you to confirm the source of the new CryoSat-2 DEM, attributed to us, used in Fig. 1 and Fig. 2 of your review? Earlier this year we made a preliminary version of the CryoSat-2 DEM available for download to the community, which is posted at a coarser resolution of 2 km and not generated with the same blended approach as described in our manuscript. As such it is a separate product from the new 1 km CryoSat-2 DEM reported in this paper. The new 1 km CryoSat-2 DEM has not yet been publicly released, but we will of course make the new dataset freely available to the scientific community in the future alongside this detailed methods paper.

[Figure]

**Figure 1: Difference between resampled ICESat DEM (WGS84 version) (DiMarzio et al., 2009) and a) new CryoSat-2 DEM presented in the submitted manuscript, b) Bamber et al. (2009) DEM (Bamber et al., 2009), and c) Helm et al. (2014) CryoSat-2 DEM (Helm et al., 2014)**

**References**

Bamber, J. L., Gomez-Dans, and J. L, Griggs, J. A.: A new digital elevation model of the Antarctic derived from combined satellite radar and laser data – Part 1: Data and methods, The Cryosphere, 3, 101-111, doi:10.5194/tc-3-101-2009, 2009.

DiMarzio, J., Brenner, A., Schutz, R., Shuman, C. A., and Zwally, H. J.: GLAS/ICESat 500m laser altimetry digital elevation model of Antarctica, Boulder, Colorado, USA. National Snow and Ice Data Center, Digital Media, 2007.

Helm, V., Humbert, A., and Miller, H.: Elevation and elevation change of Greenland and Antarctica derived from CryoSat-2, The Cryosphere, 8, 1539-1559, doi:10.5194/tc-8-1539-2014, 2014.

---

## Referee Comment (RC3) · Anonymous Referee #3 · 12 Dec 2017

Review of "A new Digital Elevation Model of Antarctica derived from CryoSat-2 altimetry" by Thomas Slater et al., submitted to The Cryosphere Discussions.

Summary: The authors present a Digital Elevation Model (DEM) of Antarctica derived from CryoSat-2 measurements acquired in the period July 2010 – July 2016. The DEM is generated from spatio-temporal fitting to elevation measurements in individual grid cells of varying size (1, 2 and 5km^2). The accuracy of the DEM is evaluated through comparison to contemporary surface measurements from NASA's Operation IceBridge campaigns.

General comment: The manuscript reads well and is easy to follow. The creation of a new DEM might not represent a major scientific step forward, but the authors argue nicely why updated and high-quality DEMs are useful for a range of applications. The

methods applied are sound and robust, and the authors find that the DEM has improved accuracy compared to previous altimetry-based DEMs of Antarctica, making this work worth publishing. I have listed below some specific recommendation that I think the authors should address to improve the manuscript.

Specific comments:

Sect 2.1.

p.3, l. 8. Both references here are from before the launch of CS-2. It would be nice to back up this statement of the performance of the OCOG with literature that actually is based on CS-2 data, if possible.

p.3, l. 9-11. It is unclear to me if the authors use the L2 SARIn product from ESA or whether they perform the retracking themselves. In the case of the latter (which I suspect is the case) some more detail about the retracking algorithm should be added, in the same way as they describe the OCOG.

Sect. 2.2.

p.3, Eq.1. The model includes a t-term taking into account any trend in surface elevation in the grid cell. But some regions must also include an acceleration ($t^2$ term). How will it affect the z value in a grid cell if an acceleration is present but not modelled?

p.3, l. 22. What is your definition of "unrealistic estimates"?

p.3, l. 29-30. The method results in 60% of the 1km grid cells on the ice sheet and 75% on the ice shelves having z values. Is this due to lack of measurements in the remaining grid cells? What is the minimum number of data that you require to provide a z-value in a grid cell? And do you have a requirement on the time span that must be covered by available measurements?

Also, have the authors performed relocation (due to topography) of the LRM data, and if so, how is this done? This should be clarified.

[Figure]

Sect. 2.4.

The approach for the DEM evaluation is well described and I think that the choice of separating the evaluation in 'observed' and 'interpolated' DEN grid cells is good. This makes the analysis and results very transparent.

Sect 3.

I think that some of the information provided in this section actually fits better under Sect 2.4 (DEM evaluation), e.g. what statistics are being derived, how to account for elevation changes etc.

p. 6, l. 16 Isn't it the case that when using Eq. 1 to find the elevation (z) in a grid cell, this elevation actually corresponds to the elevation at t=0, meaning 2010 (or whatever the start time of measurements in the grid cell is)? Did you ensure the effective time stamp to be July 2013 by defining this to be t=0 ?

Sect 3.1.

p. 6, l. 15-17 Yes, but this is only the case if the time stamp is actually July 2013 (see my previous comment).

p. 6, l. 31-33. Why didn't you correct for the temporal elevation changes here?

Figures.

Figure 6. The colour scale is not optimal. First thing is that all the differences shown all look blue/green, making it difficult to see if there are any local differences. Also, the colours of the max and min values in the colour scale look similar (to me at least). Figure 7. Same comment as for figure 6.

---

## Author Comment (AC2) · 15 Jan 2018

Thank you for your review, please find attached our response to all referee comments.

Please also note the supplement to this comment:
https://www.the-cryosphere-discuss.net/tc-2017-223/tc-2017-223-AC2-supplement.pdf

---

## Author Comment (AC3) · 15 Jan 2018

**Response to reviewers: "A new Digital Elevation Model of Antarctica derived from CryoSat-2 altimetry"**

Thomas Slater, Andrew Shepherd, Malcolm McMillan, Alan Muir, Lin Gilbert, Anna E. Hogg, Hannes Konrad, Tommaso Parrinello

We would like to thank each of the three reviewers for their positive and constructive reviews of our manuscript. In our response to each individual comment we outline where we have made changes according to suggestions which we agree improve the manuscript, or provide detailed reasoning where we have chosen to not make changes.

**Response to specific comments**

Please find below response to each of the reviewers comments in turn. Reviewer comments are in *black italic*, our responses are in blue.

**Anonymous Referee #1**

**1.** *Page 1 Line 24: "surface elevation is essential for delineation of drainage basins" – this is not totally true – although early work on drainage basins (early Rignot and Zwally papers), used surface slope to delineate drainage, later Rignot papers used flow and flow direction which is a more robust method. I would temper the sentence by saying "can be used" rather than are essential.*

Agreed, we have amended the sentence to reflect this:

"Accurate knowledge of surface elevation can be used for both the delineation of drainage basins and estimation of grounding line ice thickness, necessary for estimates of Antarctic mass balance calculated via the mass budget method (Rignot et al., 2011b; Shepherd et al., 2012; Sutterly et al., 2014)."

**2.** *Page 2 line 5: It might be worth adding that over rock outcrops and steeper slopes, where cryostat coverage is poorer, the photogrammetric models do perform better and so act as an alternative in these regions.*

Agreed, we have amended the sentence to reflect the performance of this technique over bare rock and steep slopes.

"Although these photogrammetric models perform well over regions of bare rock and steep slope found in the margins, their accuracy is considerably reduced in ice covered areas."

**3.** *Page 3 line 30. Reading the text I was a little unsure if the figures (percentages) included the polar hole or not. Please clarify.*

All stated percentages include the pole hole – after integrating the re-sampled 2 km and 5 km DEMs we have a data coverage of 94%, and we then interpolate any unobserved grid cells outside the pole hole, which accounts for 5% of the ice sheet area. This leaves 1%, which represents the area of the pole hole. We have amended this sentence to make this clearer.

"At a resolution of 1 km, the model fit provides an elevation estimate in 60 % and 75% of grid cells within the total area of the ice sheet and ice shelves, respectively."

**4.** *Page 4 line 10: Why use Kriging? There are lots of alternative interpolation methods, all will give slightly different results, but you do not say why this method was chosen rather than the others. Do you have any evidence that this is the best method to use?*

Our use of Kriging is in line with the generation of previous DEMs derived from satellite altimetry within the literature (Bamber et al., 2009; Helm et al., 2014). Ordinary kriging is a geostatistical approach which takes into account the distribution of values around the location to be interpolated, and as such is an appropriate technique for interpolating elevation. We have amended and added citations to this sentence to make this clearer:

"In order to provide a continuous dataset, we estimate elevation values in grid cells North of 88 º S that contain no data using ordinary kriging (Isaaks and Srivastava, 1989; Kitanitis 1997), an interpolation technique used in the generation of previously published DEMs (Bamber et al., 2009; Helm et al., 2014)."

**5.** *A slight grammatical point on page 7 and through the text; the phase "At the Antarctic Peninsula" and "At the Antarctic Ice shelves" should be changed to "On the..."*

Thank you for the suggestion. We have chosen to keep the use of "at the…" throughout the text, as we use the preposition "at" to indicate a specific location, which we also feel is appropriate here.

**6.** *Page 9 line 22+23: are there also more data gaps in the high slope areas that add to the inaccuracies over these regions? If so it would be worth adding the point.*

This sentence refers to the evaluation of grid cells observed by CryoSat-2 (therefore no data gaps), which when measuring high slope areas and operating in Low Resolution Mode only, can range to the peaks of undulations, thereby under-sampling troughs. When we interpolate the small proportion of grid cells that are unobserved, reduced data coverage within the search radius can lead to higher interpolation errors, which we address in Section 3.2.

**7.** *Figure 1 and figure 8: please remove the lines connecting the points as these are misrepresentative. Both these figures would be better as histograms rather than line graphs.*

Thank you for this suggestion, we have changed both of these figures to bar plots.

**8.** *In the figure legend for figure 2 I would add a point explaining the noticeable blue line where the mode changes from LRM and SARIn modes.*

Thank you for this suggestion, we have added a sentence to the figure caption:

"At the mode mask boundary, where CryoSat-2 switches between LRM and SARIn operating modes, grid cells are predominantly derived from 2 km model fits, as there are a reduced number of elevation measurements available to constrain model fits at a resolution of 1 km."

**Anonymous Referee #2**

*However, to my opinion, since the paper is presenting another Antarctic DEM a more detailed analysis and comparison to existing DEMs is required. Especially the implications, reliability of the new method compared to widely used interpolation methods is worth to investigate in more depth. E.g. a comparison with the cited CryoSat-2 DEM of Helm et.al. could be used to demonstrate if and where this new approach in combination with 6 times more data is performing better or has weaknesses. E.g. a difference plot between both CryoSat-2 DEMs over whole Antarctica would be very informative to see in which areas the DEMs differ.*

In our paper we present a comparison of our new DEM and three currently available DEMs against an independent airborne laser altimeter elevation dataset, which we feel is the most rigorous way of investigating the reliability and accuracy of each DEM. From this analysis we report a lower median and root mean squared elevation difference for the Antarctic ice sheet

when comparing our CryoSat-2 DEM than those of the other currently available DEMs, which we believe already demonstrates the accuracy and validity of our approach.

We feel that differencing two individual DEMs will illustrate their respective differences, but will not provide any information regarding their individual performance, as it is not reasonable to make an initial assumption that one DEM is more accurate than the other (which all have an average reported accuracy on the metre scale). The IceBridge dataset is an independent dataset from a different sensor with a well-documented accuracy (at the centimetre scale), and is therefore, to our opinion, an appropriate dataset with which to evaluate the DEMs. In addition, when differencing individual DEMs it is necessary to resample one DEM to the posting of the other, which can introduce interpolation errors (please see our preliminary response to your review, posted on 30th November 2017).

*I have some concerns about the applied pixel fit method. Since the elevation is generated on a pixel level (neighboring pixel are not 'talking to each other'). This might introduce elevation jumps or artefacts. As example I generated hill shades of the new DEM and the mentioned external DEMs (Fig1). The new DEM shows clearly erroneous pixels, especially in areas of steep topography, close to the Grounding line. I'm wondering if the authors can explain why this is happening and if there is a strategy to avoid this. The other DEMs do not show such artefacts.*

As we have requested in our preliminary response to your comment (posted on 30th November 2017), could you please confirm the source of our DEM used to generate the figures in your review? Earlier this year we made a preliminary version of the CryoSat-2 DEM available for download to the community, which is posted at a coarser resolution of 2 km and not generated with the same blended approach as described in our manuscript. As such it is a separate product from the new 1 km CryoSat-2 DEM reported in this paper, which is not yet publicly available. We apologise for any confusion caused by this.

We feel the accuracy of our approach at a resolution of 1 km in areas of steep topography is well demonstrated in our evaluation against IceBridge measurements (Figure 7 and Table 2 in the manuscript). In Figure 8 of our manuscript we show our DEM is more accurate in areas of steep topography (e.g. Antarctic Peninsula) than the other available DEMs.

Previous DEMs generated from smaller time intervals and using an interpolation method applied to the entire ice sheet will still likely include these artefacts as they are in the data, but are harder to spot as a result of spatial smoothing. In addition, such smoothing introduces a spatial correlation across the search radius, degrading the resolution of the DEM beyond its stated posting value. We prefer that our primary product is posted at the true resolution of the data, and therefore is a true reflection of its quality. However, we will include a supplementary smoothed version of the DEM in our final release for any users who would prefer such a product, or they can easily apply a smoothing method of their own preference.

*In table 3 a comparison of the Differences of LRM/SARIn areas to Icebridge with respect to slope is given. Since the new DEM is a composite of 'observed pixels' and 'interpolated pixels' I would like to see this analysis splitted up (similar to Table 2).*

We feel the purpose of Table 3 is to illustrate that the uncertainty of observed grid cells increases with surface slope, and is one of the principal sources of the reported elevation differences when comparing CryoSat-2 to IceBridge. While the DEM is a composite of observed and interpolated pixels, the proportion of pixels that are interpolated only represent 5% of the total Antarctic ice sheet area. These interpolated grid cells predominantly occur at the Antarctic Peninsula, Transantarctic Mountains, and the Cook Mountains – areas of rugged terrain and slope greater than 0.75° (Figure 5). The analysis with respect to slope presented in Table 3 accounts for 95% of the total ice sheet area, and gives a good depiction of the accuracy of the overall product. We believe splitting up interpolated grid cells with respect to slope in the way suggested will not add anything to the description of the accuracy of interpolated grid cells, which we feel is sufficiently addressed in Section 3.2 and Table 2.

*Furthermore, I would suggest to present two figures where the mean difference and respectively the Stddev is plotted against slope (e.g. binned to 0.05) for the whole DEM, observed and interpolated pixel, respectively. In addition, this Figure should include the same analysis for the other 3 DEMs. Such kind of figure would clearly show the difference between observed and interpolated pixels as well as the stated improvement of the new DEM against existing DEMs.*

Thank you for this suggestion, however the IceBridge airborne dataset used in our study for evaluating the accuracy of DEMs preferentially samples areas of high slope (as stated in page 6, line 25). As such, we feel binning elevation differences as a function of slope in this way will not provide a robust indication of the accuracy of a DEM, as the slope distribution of the evaluation dataset is not representative of the slope distribution of the Antarctic ice sheet.

As submitted the manuscript already contains nine figures, of which four address the accuracy of our DEM. We believe this is already a large amount of figures, and that our manuscript already demonstrates the value of our DEM when compared to existing products.

*Is there a way to compare the final slope model (Fig 5) with the 2D quadratic surface slope, estimated for each pixel?*

From a comparison of the slopes from Figure 5 in the manuscript (estimated from the dz/dx and dz/dy components) and the modelled quadratic surface slopes in observed grid cells we generally find good agreement, with a mean and RMS difference of 0.04 and 0.5 degrees, respectively for the Antarctic ice sheet. However, it should be noted that this is not a fair like-for-like comparison – the modelled quadratic surface slopes represent the slope within the pixel, whilst the slopes in Figure 5 are the calculated from pixel to pixel, so we are comparing slopes estimated over different length scales here.

We realise when we introduced the slope from the DEM in the manuscript we were ambiguous in describing how it was generated; we have amended some text to make this clearer:

"Surface slopes derived from the elevation gradient of the DEM (Fig. 5) illustrate the short scale topographic undulations, and identify the ice divides and larger features such as subglacial Lake Vostok."

*How robust is this pixel wise 2D surface fit and its sensitivity to the number and distribution of data points within a pixel? One could run the same fit method on yearly subsets of data and compare the elevations of 'observed pixels' of the subset DEMs with the Final DEM. (e.g. generate maps of the mean of the differences between subset-DEMs and final DEM and its stddv).*

While it is true that the surface fit is sensitive to the number and distribution of data points, this will apply to the accuracy of any DEM regardless of its generation method. This is why we have used use the most complete data set available here. During the generation of the DEM we removed any poorly constrained fits with a low number of data points, or where the residuals of the modelled elevation and observations are large. Performing a comparison as suggested above will not incorporate the effects of temporal change into the statistics, making it difficult to draw any conclusions. Our chosen model has been established and evaluated within the literature (Smith et al., 2009; Moholdt et al., 2010; Flament and Rémy, 2012; McMillan et al, 2014; McMillan et al., 2016; Konrad et al., 2017). We believe we have already demonstrated the validity of our approach for deriving surface elevation from our evaluation against the IceBridge airborne altimetry.

*Please make clear what kind of data you are using as input. You mention that an OCOG retracker is used for LRM data but another retracker for SARIn. Please explain why and what are the consequences (e.g bias based on the different retracker).*

We describe the input data used to generate our DEM in good detail in Section 2.1 (page 2, line 18 to page 3, line 11). In LRM and SARIn CryoSat-2 is operating in different acquisition modes: as a traditional pulse limited altimeter in LRM, and as a SAR altimeter in SARIn mode. These modes produce different waveform shapes (Wingham et al. 2006), and as such it is appropriate to treat them with different retrackers, as we have here. We believe in this manuscript it is important to address the absolute accuracy of the DEM, which we have done in detail. A relative intercomparison of different retrackers is a separate body of work beyond the scope of this manuscript.

We have added a clause in Section 2.1 which addresses the use of different retrackers:

"For the SARIn area, where CryoSat-2 operates as a SAR altimeter and waveform characteristics differ from those acquired in LRM, elevations are retrieved we using the ESA Level 2 SARIn retracker, which determines the retracking correction from fitting the measured waveform to a modelled SAR waveform (Wingham et al., 2006; ESA, 2012)."

*Are you applying any slope correction to the LRM data? If yes what method is used and which DEM is used for the slope correction? Do you use your newly derived DEM and an iterative scheme? If you use another DEM, please explain why. If you don't apply a slope correction to LRM than I see some inconsistency between the LRM and SARIn regions, since the SARIN data is slope corrected using the interferometric phase.*

The LRM measurements are slope corrected as part of the ESA Level 2 product. We provide the requested information on page 3, line 2. The slope correction in the Level 2 product is performed using an external DEM to relocate the echoing point away from nadir in accordance with the surface slope. Implementing an iterative slope correction scheme for LRM observations would require building a Level 1 processor, which is outside the scope of our study, but could be an avenue for future improvements. However, from our comparison of DEMs to the IceBridge airborne dataset (Figure 8), we demonstrate that our DEM has an improved accuracy in the Antarctic ice sheet over those that have been generated with an iterative slope correction scheme.

*Do you use all data points or do you run a pre-filtering to exclude erroneous data points before running the fit procedure? Are there any filtering approaches used after applying the pixel wise fit?*

A series of quality flags are included in the ESA L2 product, which are used for removing respective observations before the surface fit. After the fit procedure, we remove elevations from poorly constrained model fits (stated in page 3, line 22), through defining a series of quality filters in order to remove poorly constrained or unrealistic model fit solutions. These included controls on (i) the number of elevation estimates within a grid cell, and the length of the overall time series, (ii) any poorly constrained solution with a high root mean squared difference in the residuals between model fits and observations, or 1-sigma uncertainty of the elevation change estimates, and (iii) any model fit that resulted in an unrealistic elevation change or slope estimate.

We have created a new supplementary material document with a section that provides information on our filtering approach used after applying the model fit to the data, and included a reference to this supplementary material in the main text in Section 2.2.

*I also miss a map of uncertainty for whole Antarctica coming along with the DEM which would be required for ice sheet modelers.*

Thank you for the suggestion, we have generated an uncertainty map for a new supplementary material document which is derived from the RMS error of the fit in observed grid cells, and the kriging error in interpolated grid cells.

*P8 L30: Why are you not applying an elevation trend correction to the interpolated areas? This is inconsistent. You could easily generate an interpolated dhdt map to correct for elevation change in interpolated areas.*

Our estimation of surface elevation over a time period of 6 years requires consistently resolving both elevation and elevation rates from our model fit method. By definition, in areas where we don't have an elevation value, we don't have a rate of elevation change with which to correct the elevation differences. Although it would be possible to generate an interpolated dh/dt map, it would introduce further interpolation errors. Our current validation provides an upper bound of the estimate of the error in interpolated grid cells, which includes errors due to both interpolation and elevation change.

In addition, the small proportion (5%) of interpolated grid cells in our DEM predominantly occur in areas of bare rock and rugged terrain at the Antarctic Peninsula, Transantarctic Mountains, and the Cook Mountains, and not in the areas where appreciable elevation change is occurring (Figure 2 in the manuscript). We illustrate this by plotting the median difference of evaluated interpolated grid cells in ice velocity bands for the Antarctic ice sheet (please see Figure AC1), obtained from the MEaSUREs dataset (Rignot et al., 2011). In interpolated grid cells, the elevation difference is highest in slow flowing areas and generally decreases with increasing ice velocity (used here as a proxy for elevation change in the absence of a comprehensive observational dataset of elevation change, which would of course be more appropriate). As such, when evaluating interpolated grid cells against IceBridge, we believe the elevation difference is mainly due to the interpolation error, and not due to the temporal difference between the two estimates.

We have amended the text in the manuscript to make this clearer:

"We note that, because the elevation rate is unknown where there is no model solution, we have not corrected for temporal changes in elevation between the acquisition periods of the two datasets within our evaluation of interpolated grid cells. As a result, the reported values represent an upper bound of the elevation difference which includes errors due to both interpolation and elevation change — if present within an interpolated grid cell."

[Figure]

**Figure AC1: Median difference of interpolated DEM grid cells in the Antarctic ice sheet evaluated against IceBridge airborne measurements, binned into ice velocity bands 20 m/yr in size. 90% of all interpolated DEM grid cells in the Antarctic ice sheet are located in regions flowing slower than 340 m/yr, shown in the vertical dashed line.**

*Fig 8: Why is the median difference below zero in all cases - this would mean that CryoSat is measuring above the laser surface?*

All presented elevation differences are DEM - IceBridge (stated in page 5, line 22), therefore a negative elevation difference indicates that CryoSat-2 is measuring below the laser surface. This is within expectations – the Ku band operating frequency of CryoSat-2 will penetrate beyond a dry ice sheet surface into the snow pack, whereas the laser altimeter on-board IceBridge survey craft will return from the air-snow interface.

*Figures (2,3,4,5,6,7,9): please use km instead of m and overplot a Latitude / Longitude grid on top of the polarstereo projection.*

Thank you for this suggestion, we have added a latitude and longitude grid to the figures.

*Furthermore, I think that the new DEM is shifted by 1 pixel. Difference plots as suggested above show a strange pattern (see Fig. 2) which is not observed between the other 3 DEMs - Do you have any explanation?*

Please see our preliminary response to your review posted on 30th November 2017 regarding this comment, where we have addressed this in more detail.

**Anonymous Referee #3**

**Sect 2.1**

*p.3, l. 8. Both references here are from before the launch of CS-2. It would be nice to back up this statement of the performance of the OCOG with literature that actually is based on CS-2 data, if possible.*

Thank you for this suggestion, we have also added a reference which highlights the use of the OCOG algorithm with CryoSat-2 data.

*p.3, l. 9-11. It is unclear to me if the authors use the L2 SARIn product from ESA or whether they perform the retracking themselves. In the case of the latter (which I suspect is the case) some more detail about the retracking algorithm should be added, in the same way as they describe the OCOG.*

Yes, we have used the ESA L2 SARIn product, please see Section 2.1 (page 2, line 18). Accordingly, we include the relevant citations which describe the algorithm in full detail, if the reader wishes to know more (page 3, line 10). We have also amended some wording in Section 2.1 to make clear the L2 product includes retracked elevations:
"Within the LRM mode mask area we select Level 2 elevation estimates retrieved using the Offset Centre of Gravity (OCOG) retracking algorithm (Wingham et al., 1986), which defines a rectangular box around the centre of gravity of an altimeter waveform based upon its power distribution (Wingham et al., 1986)."

"For the SARIn area, where CryoSat-2 operates as a SAR altimeter and waveform characteristics differ from those acquired in LRM, elevations are retrieved we using the ESA Level 2 SARIn retracker, which determines the retracking correction from fitting the measured waveform to a modelled SAR waveform (Wingham et al., 2006; ESA, 2012)."

**Sect 2.2**

*p.3, Eq.1. The model includes a t-term taking into account any trend in surface elevation in the grid cell. But some regions must also include an acceleration (tˆ2 term). How will it affect the z value in a grid cell if an acceleration is present but not modelled?*

For our DEM elevation we select the mean elevation parameter, $\bar{z}$, which represents the elevation at the centre of our chosen time window, which will be insensitive to higher order terms in time. Our approach, which models the surface evolution over time, represents an evolution in the generation of DEMs, allowing the use of longer observation periods.

In addition, recorded regions where acceleration has occurred are rare in Antarctica (Konrad et al., 2017) – the surface elevation of the majority of the continent is either not changing, or changing linearly with time. Any acceleration over our 6-year observation period will not be occurring fast enough to warrant accounting for in our modelled solution. Our chosen model is well established and evaluated in the literature to accurately capture elevation change (Smith et al., 2009; Moholdt et al., 2010; Flament and Rémy, 2012; McMillan et al, 2014; McMillan et al., 2016). Finally, adding more terms will make our model fit more unstable. Because of this, and the points above, we feel our use of a linear rate of elevation change is appropriate here.

*p.3, l. 22. What is your definition of "unrealistic estimates"?*

This comment was also raised by Anonymous Referee #2, please find our response duplicated below:

A series of quality flags are included in the ESA L2 product, which are used for removing respective observations before the surface fit. After the fit procedure, we remove elevations from poorly constrained model fits (stated in page 3, line 22), through defining a series of quality filters in order to remove poorly constrained or unrealistic model fit solutions. These included controls on (i) the number of elevation estimates within a grid cell, and the length of the overall time series, (ii) any poorly constrained solution with a high root mean squared difference in the residuals between model fits and observations, or 1-sigma uncertainty of the elevation change estimates, and (iii) any model fit that resulted in an unrealistic elevation change or slope estimate.

We have created a new supplementary material document with a section that provides information on our filtering approach used after applying the model fit to the data, and included a reference to this supplementary material in the main text in Section 2.2.

*p.3, l. 29-30. The method results in 60% of the 1km grid cells on the ice sheet and 75% on the ice shelves having z values. Is this due to lack of measurements in the remaining grid cells? What is the minimum number of data that you require to provide a z-value in a grid cell? And do you have a requirement on the time span that must be covered by available measurements?*

Correct, elevations were derived from 2 km and 5 km model fits when there was an insufficient number of elevation measurements to accurately constrain model fits at the finer grid cell resolutions of 1 km and 2 km, respectively. We discarded any model fit elevations that were derived from less than 15 measurements, or a time series of less than 2 years in length. This information is now available in the previously mentioned supplementary material document, where we describe our data filtering approach in more detail.

*Also, have the authors performed relocation (due to topography) of the LRM data, and if so, how is this done? This should be clarified.*

The LRM measurements are slope corrected as part of the ESA Level 2 product, please see page 3, line 2. The slope correction for LRM measurements is performed using an external DEM to relocate the echoing point in accordance with the slope. More information is available to the reader through the cited literature.

**Sect 3**

*I think that some of the information provided in this section actually fits better under Sect 2.4 (DEM evaluation), e.g. what statistics are being derived, how to account for elevation changes etc.*

Thank you for your suggestion – we feel this information is equally valid in either section, and have chosen to keep it in Section 3. Because the majority of the results reported are in terms of these statistics, we believe it is appropriate to include it in Section 3, in terms of making it easier to follow for the reader. When accounting for elevation changes, we only do so in observed regions, so we feel this information is most appropriate in the section which addresses observed grid cells.

*p. 6, l. 16 Isn't it the case that when using Eq. 1 to find the elevation (z) in a grid cell, this elevation actually corresponds to the elevation at t=0, meaning 2010 (or whatever the start time of measurements in the grid cell is)? Did you ensure the effective time stamp to be July 2013 by defining this to be t=0?*

When forming the DEM we do not use the modelled elevation Z, we use the mean elevation parameter $\bar{z}$, which is representative of the elevation at the midpoint of the elevation time series. We believe that Table 1 and Figure 6 evidence that the DEM timestamp is indeed July 2013. We have also adjusted equation 1 and amended the text to make it clear to the reader that $\bar{z}$ corresponds to the elevation at July 2013, which is the midpoint of our observation period:

"We form the DEM from the mean elevation term, $\bar{z}$, in Eq. (1) within each 1 x 1 km grid cell, which corresponds to the elevation at the midpoint of our observation period."

**Sect 3.1**

*p. 6, l. 15-17 Yes, but this is only the case if the time stamp is actually July 2013 (see my previous comment).*

Please see our response to your previous comment regarding the effective time stamp of the DEM.

*p. 6, l. 31-33. Why didn't you correct for the temporal elevation changes here?*

Please note in addressing this comment we are assuming you are referring to page 8, lines 31-33 in Section 3.2, and our choice to not correct for elevation change when evaluating interpolated grid cells. This was also raised by Anonymous Referee #2, please find our response duplicated below:

Our estimation of surface elevation over a time period of 6 years requires consistently resolving both elevation and elevation rates from our model fit method. By definition, in areas where we don't have an elevation value, we don't have a rate of elevation change with which to correct our elevation differences. Although it would be possible to generate an interpolated dh/dt map, it would introduce further interpolation errors. Our current validation provides an upper bound of the estimate of the error in interpolated grid cells, which includes errors due to both interpolation and elevation change.

In addition, the small proportion (5%) of interpolated grid cells in our DEM predominantly occur in areas of bare rock and rugged terrain at the Antarctic Peninsula, Transantarctic Mountains, and the Cook Mountains, and not in the areas where appreciable elevation change is occurring (Figure 2 in the manuscript). We illustrate this by plotting the median difference of evaluated interpolated grid cells in ice velocity bands for the Antarctic ice sheet (please see Figure AC1), obtained from the MEaSUREs dataset (Rignot et al., 2011). In interpolated grid cells, the elevation difference is highest in slow flowing areas and generally decreases with increasing ice velocity (used here as a proxy for elevation change in the absence of a comprehensive observational dataset of elevation change, which would of course be more appropriate). As such, when evaluating interpolated grid cells against IceBridge, we believe the elevation difference is mainly due to the interpolation error, and not due to the temporal difference between the two estimates.

We have amended the text in the manuscript to make this clearer:

"We note that, because the elevation rate is unknown where there is no model solution, we have not corrected for temporal changes in elevation between the acquisition periods of the two datasets within our evaluation of interpolated grid cells. As a result, the reported values represent an upper bound of the elevation difference which includes errors due to both interpolation and elevation change — if present within an interpolated grid cell."

**Figures**
*Figure 6. The colour scale is not optimal. First thing is that all the differences shown all look blue/green, making it difficult to see if there are any local differences. Also, the colours of the max and min values in the colour scale look similar (to me at least).*
*Figure 7. Same comment as for figure 6.*

Agreed, thank you for this suggestion. We have changed the colour scale for the difference plots to a divergent one, which we hope makes them easier to read.

[revised manuscript text omitted]

**1 Elevation retrieval from CryoSat-2: data filtering approach**

After the application of Eq. (1) to our selected CryoSat-2 dataset, we apply a series of quality filters to remove unrealistic elevation estimates from poorly constrained model fits. In order to identify these unrealistic elevation estimates, we introduce constraints based upon (i) data availability, (ii) data quality and (iii) physical plausibility (Table S1). Controls on data availability include a minimum number of data points, and length of the elevation time series within each grid cell. In each case, threshold values have been tested and found to be optimal. A minimum of 15 data points, and time series length of at least 2 years ensures that each model fit is constrained by sufficient data. We remove poorly constrained fits by introducing thresholds on the goodness of fit, with maximum values of the root mean squared difference of the elevation residuals from the model fit, and 1-sigma uncertainty in the elevation rate being 10 m and 0.4 m $yr^{-1}$, respectively. Finally, we remove any estimates that are not physically plausible. Maximum rates of elevation change have been reported to be 9 m $yr^{-1}$ in Antarctica (McMillan et al., 2014), so we remove any elevation rate estimate with a magnitude exceeding 10 m $yr^{-1}$. In addition, we impose a maximum quadratic surface slope value of 5 ° to maintain physically plausible glacier driving stresses — averaged over the grid cell resolution — which are controlled by the slope.

After data filtering, the model fit provides an elevation estimate for 60 %, 91 % and 94 % of grid cells within the Antarctic ice sheet at resolutions of 1 km, 2 km and 5 km, respectively. For the Antarctic ice shelves, 75 % and 98 % of grid cells at resolutions of 1 km and 2 km, respectively, contain an elevation estimate from our model fit approach after quality filtering has been applied.

| Parameter | Selection criteria |
|---|---|
| Number of data points | ≤ 15 |
| Time series length (yrs) | ≤ 2 |
| Root mean squared difference of elevation residuals from model fit (m) | ≥ 10 |
| 1-sigma uncertainty in dh/dt (m yr$^{-1}$) | ≥ 0.4 |
| \|dh/dt\| (m yr$^{-1}$) | ≥ 10 |
| Surface slope (°) | ≥ 5 |

**Table S1: Selection criteria used to remove elevation estimates resulting from poorly constrained model fits.**

2 Uncertainty map

[Figure]

**Figure S1: Uncertainty map of the new CryoSat-2 Antarctic DEM, calculated from the root mean squared difference of elevation residuals in observed grid cells, and the kriging variance error in interpolated grid cells.**

**References**

McMillan, M., Shepherd, A., Sundal, A., Briggs, K., Muir, A., Ridout, A., Hogg, A., and Wingham, D.: Increased ice losses from Antarctica detected by CryoSat-2, Geophys. Res. Lett., 41, 3899–3905, doi:10.1002/2014GL060111, 2014GL060111, 2014.

---

## Author Response (AR2)

**Response to editor request for minor revisions: "A new Digital Elevation Model of Antarctica derived from CryoSat-2 altimetry"**

Thomas Slater, Andrew Shepherd, Malcolm McMillan, Alan Muir, Lin Gilbert, Anna E. Hogg, Hannes Konrad, Tommaso Parrinello

Dear Bert,

Thank you for your response to our revised manuscript. We have addressed each of your suggestions below, with our responses written in blue. These are followed by a marked up manuscript which illustrates the changes.

1. Mention in the introduction that this version of the DEM supercedes the previously distributed preliminary version

Thank you for this suggestion, we agree it is important to make this clear for users of the DEM. We have added the following text at the end of the introduction:

*"The DEM we describe here features several improvements over the preliminary ESA CryoSat-2 Antarctic DEM distributed in March 2017, and should be used in its place. These improvements include an increase in resolution from 2 km to 1 km, an increase in data coverage on the grounded ice sheet from 91 % to 94 %, and the use of a more robust ordinary kriging interpolation scheme to provide a continuous elevation dataset."*

2. Uncertainties are an essential component of any data set, as important as the reported data itself. The uncertainty map in the supplement should be moved to the main manuscript, either as a separate figure, or included in fig. 4. Uncertainties should be provided for the slope map as well. In order to reduce the number of figures, fig. 2 and/or 3 can be moved to the supplement.

We have changed the figures which presented the DEM and slope to 2 panel figures, which include the estimated uncertainty for each dataset in the second panel.

3. Clearly explain how the backscatter correction was applied.

We have added a new section to the supplementary which explains how the backscatter correction is calculated, and added text in the main manuscript to indicate this to the reader:

*"The elevation change trends were corrected for temporal fluctuations in backscattered power, which can introduce spurious signals in time series of elevation change, by adjusting the elevation time series according to the correlation between changes in elevation and backscattered power (see Sect. 2 of supplementary material) (Wingham et al., 1998; Davis and Ferguson, 2004; Khvorostovsky, 2012)."*

*"2 Backscatter correction to elevation change time series*
*We correct for temporally correlated fluctuations in changes in elevation and radar backscatter, known to introduce spurious signals in time series of elevation obtained using satellite radar altimetry (Wingham et al., 1998; Davis and Ferguson, 2004;*

*Khvorostovsky, 2012). To apply this correction (Eq. (S1)), we first compute the correlation gradient in elevation as a function of power, dh/dp, using a linear fit in each grid cell over a 60-month time window. We then multiply time series of changes in backscattered power dp by this gradient to estimate the backscatter correction term, which we remove from our original elevation change time series dh.*

$$dh_{corrected} = dh - (dp\frac{dh}{dp}) \qquad (S1)"$$

4. Explicitly mention that in the LRM mode, an external DEM was used for the slope correction, and which one. The reader should not have to dig into the referenced document to find this out.

We have added the following text to make this clear to the reader:

*"As part of the Level 2 processing chain, elevation measurements recorded in LRM are slope corrected by relocating the echoing point away from nadir in accordance with the surface slope, determined using an external DEM (Radarsat Antarctic Mapping Project version 2 DEM, posted at 200 m) (Liu et al., 2001; ESA 2012). SARIn acquisitions are slope corrected using the interferometric phase difference calculated at the location of the elevation measurement."*

5. In the caption of the uncertainty map, mention that uncertainties due to (remaining) firn penetration of the radar signal are not included.

We have added this explanation into the figure caption:

*"Figure 4: a) A new elevation model of Antarctica derived from 6 years of CryoSat-2 radar altimetry data acquired between July 2010 and July 2016, and b) uncertainty map of the new CryoSat-2 Antarctic DEM, calculated from root mean square difference of elevation residuals in observed grid cells, and the kriging variance error in interpolated grid cells. Uncertainties due to radar penetration into a dry snowpack are not accounted for."*

6. In the supplement, include the difference map between your DEM and the other three DEMs mentioned in the manuscript, as suggested with R2. I agree with you that this is not necessarily an indicator of the quality of the new DEM, but it will give potential users of your DEM an idea of where to expect differences.

We have included such a figure in the supplement, which replaces the uncertainty map in Figure S1 which has now been moved to the main manuscript.

7. Include a figure showing mean difference/standard deviation against slope, as suggested by R2. I agree that the IceBridge data may not be representative for the entire AIS slope distribution, and it might be good to add a caveat about this in the discussion of the figure.

We have added such a figure as Figure 10 in the manuscript, where we have binned median and RMS differences into slope bins 0.05 ° in size, as suggested by Reviewer #2. We have also added the following text which discusses the figure in section 3.3:

*"Similarly, median and root mean squared differences calculated with respect to surface slope for each DEM within the Antarctic ice sheet (Fig. 10), further illustrate the improvement offered by the new CryoSat-2 DEM, and the slope-dependency*

*of DEM accuracy. We limit this analysis to regions where surface slopes are lower than 1.5 °, which accounts for approximately 94 % of the Antarctic ice sheet area North of 88 ° S. In addition, we note that the spatial distribution of the airborne dataset used for comparison within the grounded ice sheet preferentially samples regions of high slope and low elevation, and does not reflect the overall elevation and slope distributions of the Antarctic ice sheet. Approximately 60 % of DEM grid cells overflown by IceBridge survey craft have an elevation of less than or equal to 1000 m, and 43 % have a surface slope greater than 0.5 °. In comparison, approximately 15 % and 22 % of the Antarctic ice sheet area has elevations of less than 1000 m and slopes greater than 0.5 °, respectively."*

R2 would like to look further into the pixel shift observed in the preliminary DEM and like to have access to the current DEM. Given that your manuscript is mainly a discussion of a new data set, I think it is a reasonable request for the reviewers to have access to the data. Of course, it will be made clear that the data can only be used for reviewing purposes. Please send me a link to a download location for the new DEM, which I will forward to R2.

*We hope this issue has been resolved as a result of our previous communication, and that Reviewer #2 is no longer experiencing this issue with the preliminary or new DEM version. We have added a product format description into the supplement, which makes clear that x and y coordinates refer to the centre of the grid cell, in the hope that this issue will not be encountered by other users.*

*"4 Product format*

- *DEM and supplementary grids (slope, grid cell data composition, uncertainty) available in netcdf and geotiff formats*
- *Projection: Polar Stereographic*
- *Ellipsoid: WGS-84*
- *Standard latitude: -71 °*
- *Central meridian: 0 °*
- *Minimum x value: -2819500 m*
- *Minimum y value: -2419500 m*
- *Grid spacing: 1000 m*
- *Coordinate convention: Grid coordinate values relate to the centre of each grid cell*
- *Number of rows: 4840*
- *Number of columns: 5640*
- *DEM units: metres*
- *Data format: big-endian IEEE 4-byte floating point format, no data value is NaN"*

[revised manuscript text omitted]

**1 Elevation retrieval from CryoSat-2: data filtering approach**

After the application of Eq. (1) to our selected CryoSat-2 dataset, we apply a series of quality filters to remove unrealistic elevation estimates from poorly constrained model fits. In order to identify these unrealistic elevation estimates, we introduce constraints based upon (i) data availability, (ii) data quality and (iii) physical plausibility (Table S1). Controls on data availability include a minimum number of data points, and length of the elevation time series within each grid cell. In each case, threshold values have been tested and found to be optimal. A minimum of 15 data points, and time series length of at least 2 years ensures that each model fit is constrained by sufficient data. We remove poorly constrained fits by introducing thresholds on the goodness of fit, with maximum values of the root mean squared difference of the elevation residuals from the model fit, and 1-sigma uncertainty in the elevation rate being 10 m and 0.4 m yr$^{-1}$, respectively. Finally, we remove any estimates that are not physically plausible. Maximum rates of elevation change have been reported to be 9 m yr$^{-1}$ in Antarctica (McMillan et al., 2014), so we remove any elevation rate estimate with a magnitude exceeding 10 m yr$^{-1}$. In addition, we impose a maximum quadratic surface slope value of 5 ° to maintain physically plausible glacier driving stresses — averaged over the grid cell resolution — which are controlled by the slope.

After data filtering, the model fit provides an elevation estimate for 60 %, 91 % and 94 % of grid cells within the Antarctic ice sheet at resolutions of 1 km, 2 km and 5 km, respectively. For the Antarctic ice shelves, 75 % and 98 % of grid cells at resolutions of 1 km and 2 km, respectively, contain an elevation estimate from our model fit approach after quality filtering has been applied.

| Parameter | Selection criteria |
|---|---|
| Number of data points | ≤ 15 |
| Time series length (yrs) | ≤ 2 |
| Root mean squared difference of elevation residuals from model fit (m) | ≥ 10 |
| 1-sigma uncertainty in dh/dt (m yr$^{-1}$) | ≥ 0.4 |
| \|dh/dt\| (m yr$^{-1}$) | ≥ 10 |
| Surface slope (°) | ≥ 5 |

**Table S1: Selection criteria used to remove elevation estimates resulting from poorly constrained model fits.**

**2 Backscatter correction to elevation change time series**

We correct for temporally correlated fluctuations in changes in elevation and radar backscatter, known to introduce spurious signals in time series of elevation obtained using satellite radar altimetry (Wingham et al., 1998; Davis and Ferguson, 2004; Khvorostovsky, 2012). To apply this correction (Eq. (S1)), we first compute the correlation gradient in elevation as a function of power, $\frac{dh}{dp}$, using a linear fit in each grid cell over a 60-month time window. We then multiply time series of changes in backscattered power $dp$ by this gradient to estimate the backscatter correction term, which we remove from our original elevation change time series $dh$.

$$dh_{corrected} = dh - (dp\frac{dh}{dp}) \qquad \text{(S1)}$$

**2 DEM difference maps**

[Figure]

a) Bamber et al., 2009   b) Bedmap2 (Fretwell et al., 2013)   c) Helm et al., 2014

**Figure S1:** **Maps of differences calculated between the new CryoSat-2 DEM described in this manuscript, and the three publically available DEMs used in Figure 8 (Bamber et al., 2009; Fretwell et al,. 2013; Helm et al., 2014). In each case the previously published DEM is resampled to the posting of the new CryoSat-2 DEM, and differences calculated as the resampled DEM subtracted from the new CryoSat-2 DEM.**

**4 Product format**

- DEM and supplementary grids (slope, grid cell data composition, uncertainty) available in netcdf and geotiff formats
- Projection: Polar Stereographic
- Ellipsoid: WGS-84
- Standard latitude: -71 °
- Central meridian: 0 °
- Minimum x value: -2819500 m
- Minimum y value: -2419500 m
- Grid spacing: 1000 m
- Coordinate convention: Grid coordinate values relate to the centre of each grid cell
- Number of rows: 4840
- Number of columns: 5640
- DEM units: metres
- Data format: big-endian IEEE 4-byte floating point format, no data value is NaN